# Heat stress promotes longevity in budding yeast by relaxing the confinement of age-promoting factors in the mother cell

Sandro Baldi[†‡], Alessio Bolognesi[†], Anne Cornelis Meinema[†], Yves Barral[*]

Institute of Biochemistry, Department of Biology, ETH Zürich, Zürich, Switzerland

**Abstract** Although individuals of many species inexorably age, a number of observations established that the rate of aging is modulated in response to a variety of mild stresses. Here, we investigated how heat stress promotes longevity in yeast. We show that upon growth at higher temperature, yeast cells relax the retention of DNA circles, which act as aging factors in the mother cell. The enhanced frequency at which circles redistribute to daughter cells was not due to changes of anaphase duration or nuclear shape but solely to the downregulation of the diffusion barrier in the nuclear envelope. This effect depended on the PKA and Tor1 pathways, downstream of stress-response kinase Pkc1. Inhibition of these responses restored barrier function and circle retention and abrogated the effect of heat stress on longevity. Our data indicate that redistribution of aging factors from aged cells to their progeny can be a mechanism for modulating longevity.
DOI: https://doi.org/10.7554/eLife.28329.001

*For correspondence:
yves.barral@bc.biol.ethz.ch

†These authors contributed equally to this work

Present address: ‡Biomedical Center Munich, Molecular Biology, Ludwig Maximilian University of Munich, Munich, Germany

Competing interests: The authors declare that no competing interests exist.

## Introduction

Many cell types divide asymmetrically to generate a naive daughter cell that renews the division potential of the lineage, and a committed daughter cell that progresses toward differentiation and generally shows a limited division potential (*Knoblich, 2010*; *Chen et al., 2016*; *Ouellet and Barral, 2012*). This is the case for many stem cells, which have the dual function of maintaining an eternal division potential and of generating differentiating daughters that eventually integrate themselves both structurally and functionally into organs (*Chen et al., 2016*; *Fisher and Sozzani, 2016*). Accordingly, while the stem cell remains young during most of the lifespan of the individual, differentiating daughters age and need to be replaced over time. Current knowledge indicates that aging in metazoans follows a progressive loss of stem cells proliferation, and hence a loss in the regeneration potential of the organs (*Van Zant and Liang, 2003*; *Beerman and Rossi, 2015*; *Ahmed et al., 2017*). However, how aging takes place at the cellular level is not well understood. Particularly, we do not yet understand how asymmetric division generates both one naive but rejuvenated and one committed but aging daughter cell.

The unicellular fungus *Saccharomyces cerevisiae* is an excellent model for studying this process (*Higuchi-Sanabria et al., 2014*; *Denoth Lippuner et al., 2014*). Indeed, these cells proliferate through budding small, rejuvenated daughter cells from the surface of the larger, mother cell (*Mortimer and Johnston, 1959*; *Hartwell and Unger, 1977*; *Kennedy et al., 1994*; *Henderson and Gottschling, 2008*). Strikingly, with each daughter produced, the mother cell ages and progressively loses its division potential until it eventually stops proliferating and dies. This process is called replicative aging and the replicative lifespan, that is, the number of daughters a mother cell generates before dying, is limited, reaching about 25 generations in average for haploid wild-type cells (*Henderson and Gottschling, 2008*; *Denoth Lippuner et al., 2014*). Beyond limiting the

**eLife digest**  Aging is often an inevitable part of life. Yet, it seems like the aging process can be sped up or slowed down in organisms as distinct as fruit flies, mice and budding yeast in response to changes in the environment. Yeast, for example, lives 40% longer when grown at the elevated – and mildly stressful – temperature of 37°C instead of 30°C. But how can yeast, or any other organism, change its lifespan in response to stressful conditions?

Budding yeast (*Saccharomyces cerevisiae*), as its name suggests, divides by budding small daughter cells from its surface. The mother cell gets older with each division, whereas the age of each daughter cell is reset to zero. The mother cell protects the daughters by keeping some harmful aging factors for itself. Many aging factors – like toxic DNA circles – are anchored to the membranes of the endoplasmic reticulum (the compartment in the cell where many proteins are made) and the nucleus (the compartment where the cell's genetic information is stored). Before the cells divide, a diffusion barrier keeps molecules in the membranes of the mother cell, preventing them from entering the membranes of the daughter cell.

The aging factors only leak into the daughter cells if they detach from the membrane in the mother or if the diffusion barrier becomes permeable. The loss of aging factors causes the mother cell to live much longer than normal. Baldi, Bolognesi, Meinema et al. asked if something similar occurred when cells experience stress, which could explain why stressed yeast cells live longer.

Indeed, mother cells did redistribute toxic DNA circles to their daughters when grown at a higher temperature. This did not happen because the DNA circles detached from the membrane. Instead the diffusion barrier became more permeable. Baldi et al. then went on to show that it was not just the heat that weakened the barrier. Rather the diffusion barrier was specifically down-regulated by one of the yeast's normal responses to stress. Lastly, when Baldi et al. took steps to make the diffusion barrier in mildly stressed cells less permeable again, the cells largely resumed aging like unstressed cells.

Together these data suggest that yeast cells undergoing mild stress might not repair damage or clear out the aging factors but rather dispose of the factors by passing them on to their offspring. It is possible that this helps the population to cope with the stress, by sharing the burden of age – or, as Baldi et al. also discuss, the wisdom of age – with the other individuals.

Stress-response pathways are conserved among many other organisms, and similar diffusion barriers occur in worms and mammals too. Thus, this newly discovered process might also happen in other cells that divide asymmetrically, including human stem cells.

DOI: https://doi.org/10.7554/eLife.28329.002

lifespan, yeast aging also manifests itself through a number of additional traits, such as the formation of protein aggregates (*Aguilaniu et al., 2003*; *Erjavec et al., 2007*; *Hill et al., 2014*; *Saarikangas and Barral, 2015*), the neutralization of the vacuolar pH (*Hughes and Gottschling, 2012*; *Henderson et al., 2014*), the fractionation of mitochondrial organization (*Hughes and Gottschling, 2012*) and the decreased sensitivity of the cell to signaling pheromone (*Smeal et al., 1996*; *Caudron and Barral, 2013*; *Schlissel et al., 2017*) reviewed in *Denoth Lippuner et al. (2014)*. In contrast, the daughter cells reset their vacuolar pH, mitochondrial organization, phero-mone response and division potential. They then become mother cells themselves; they start bud-ding-off daughters and aging.

The progressive decline of cellular fitness with age is thought to be driven by the retention and accumulation of so-called aging factors in the mother cell. Three types of aging factors have been described. First, plasma-membrane proteins such as the proton-exporter Pma1 and several multi-drug transporters remain in the mother cell as it divides and contribute to its fitness decay (*Eldakak et al., 2010*; *Henderson et al., 2014*; *Thayer et al., 2014*). Second, aging yeast mother cells also form a deposit that accumulates protein aggregates (*Aguilaniu et al., 2003*; *Erjavec et al., 2007*; *Hill et al., 2014*; *Saarikangas and Barral, 2015*). Cells that fail to form this aggregate are long-lived (*Hill et al., 2014*; *Saarikangas and Barral, 2015*). Third, intra-chromosomal recombination between repeated rDNA units excise extrachromosomal rDNA circles (ERCs) that segregate to and accumulate in the mother cell nucleus (*Szostak and Wu, 1980*; *Sinclair and Guarente, 1997*;

*Shcheprova et al., 2008*). Except for the endogenous two micron plasmid, ERCs and actually all DNA circles tested so far accumulate in the mother cell with age and accelerate aging (*Murray and Szostak, 1983*; *Falcón and Aris, 2003*). Old mother cells contain up to thousand ERCs and this load, which increases exponentially with successive divisions, might be what ultimately kills the cell (*Sinclair and Guarente, 1997*). High-fidelity retention in the mother cell of the DNA circles and of the precursors of protein aggregation is facilitated by the formation of lateral diffusion barriers in the ER membrane and the outer nuclear membrane at the bud neck (*Luedeke et al., 2005*; *Shcheprova et al., 2008*; *Clay et al., 2014*; *Saarikangas et al., 2017*). These barriers limit exchange of membrane-proteins between mother and bud. Therefore, retention of aging factors in the mother cell relies on their anchorage into the ER-membrane. Retention of the aggregation precursors relies on their membrane attachment through the farnesylated chaperone Ydj1 (*Saarikangas et al., 2017*). DNA circles attach to the nuclear envelope through the SAGA complex and nuclear pore complexes (NPCs) (*Shcheprova et al., 2008*; *Denoth-Lippuner et al., 2014*).

Remarkably, yeast cells show an extended life span when subjected to mild stresses such as calorie restriction and growth at 37°C (*Shama et al., 1998a*; *Shama et al., 1998b*; *Swieciło et al., 2000*; *Kapahi et al., 2017*). Similar effects take place in organisms as distinct as nematodes, flies and mice, indicating that the regulation of longevity involves similar regulatory pathways in all these organisms, at least upon calorie restriction, namely the TOR and PKA pathways (*Steinkraus et al., 2008*; *Kapahi et al., 2010*; *Wasko and Kaeberlein, 2014*). How these regulatory pathways actually modulate ageing progression itself is largely unknown.

The fact that yeast cells are able to modulate their longevity in response to environmental signals suggests that they have some control on the generation and accumulation of aging factors, or on the impact that these have on the physiology of the cell. We reasoned that one potential mechanism for increasing longevity could be the down-regulation of the retention of aging factors in the mother cell. Indeed, mutants affecting the retention of DNA circles in the mother cell are long-lived (*Shcheprova et al., 2008*; *Denoth-Lippuner et al., 2014*). Thus, we set here out to test whether physiological stresses affect the retention of aging factors in the mother cell, and how.

## Results

### The confinement of DNA circles is reduced upon heat stress

In order to investigate whether the confinement of aging factors in the mother cell is affected upon conditions that promote longevity, we asked whether a model DNA circle (*Shcheprova et al., 2008*; *Denoth-Lippuner et al., 2014*) was more likely to propagate to the bud upon heat stress or calorie restriction than under optimal growth conditions. This model DNA circle carries a centromere flanked with LoxP sites and hence, is turned into a non-centromeric circle upon expression of the Cre-recombinase and excision of the centromeric sequence. It also carries an array of repeated TetO sequences. Expression of the protein TetR, which binds to the TetO sequence, fused to GFP (TetR-GFP) allow the visualization of the circle as a fluorescent dot in vivo. An autonomously replicating sequence (ARS) promotes the replication of the DNA circle during S-phase. We transfected this model circle into cells co-expressing Cre fused to an estradiol-binding domain (Cre-EBD) and TetR-GFP. EBD mediates the retention of Cre in the cytoplasm until β-Estradiol is added to the growth medium (*Lindstrom and Gottschling, 2009*). Upon β-Estradiol treatment and consequent centromere excision (see Materials and methods for details) the non-centromeric DNA circles detaches from the Spindle (Spc42-CFP marks the spindle pole bodies - SPB, *Figure 1A,B*), and no-longer segregate symmetrically, unlike their centromeric counterparts. Using this system, we asked whether cells grown in conditions of calorie restriction (0.1% glucose, 30°C) or heat stress (2% glucose, 37°C) affected the retention of the circles in the mother cell compared to cells maintained in optimal growth conditions (2% glucose, 30°C). In cells grown under optimal conditions, the circles passed very infrequently to the bud (frequency of propagation to the bud: $0.04 \pm 0.01$). Although calorie restriction had no effect (propagation frequency: $0.04 \pm 0.01$), the frequency at which individual plasmids passed to the bud was increased four folds in cells grown at 37°C (propagation frequency: $0.17 \pm 0.01$). This value is very similar to what we observed in cells lacking the diffusion barrier in the outer nuclear membrane (*bud6Δ* mutant cells, propagation frequency: $0.13 \pm 0.01$; *Shcheprova et al. (2008)*; *Denoth-Lippuner et al., 2014*; *Figure 1B,C*). Thus, these data suggested

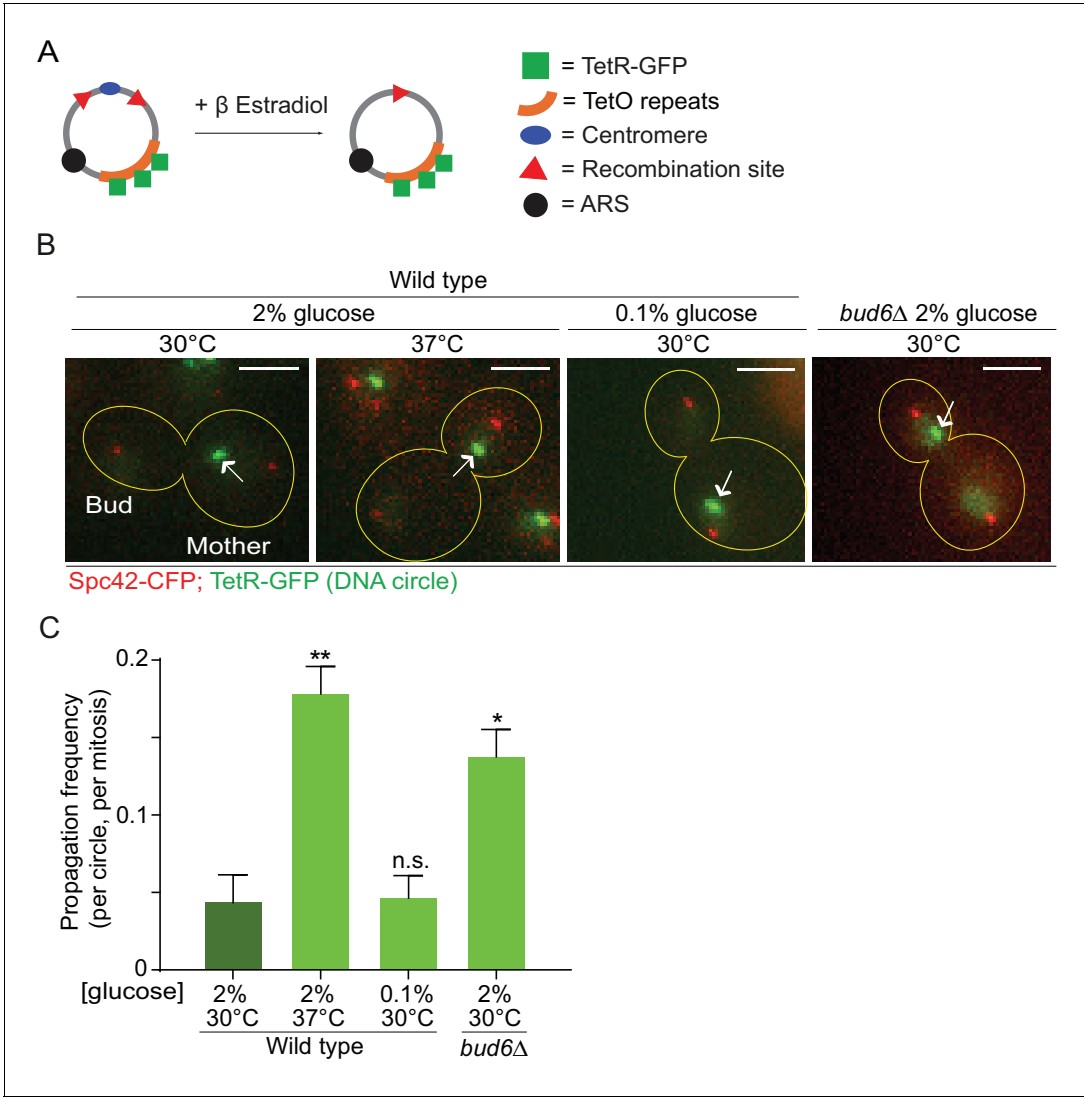

**Figure 1.** Heat stress reduces the confinement of DNA circles in the mother cell upon mitosis. (**A**) Model DNA circle with excisable centromere (blue) and TetO repeats (orange), which can be labeled with TetR-GFP (green) for visualization. ARS is an autonomously replicating sequence (black). (**B**) Examples of anaphase cells (maximum intensity projection, outline of the cell in yellow, scale bar is 3 μm). The arrows indicate DNA circles. The SPBs appear in red due to image processing. (**C**) Propagation frequency: mean ± SEM of three experiments (six for calorie restriction) with $31 \leq n \leq 122$ cells per experiment; unpaired t-test, **p<0.01.

DOI: https://doi.org/10.7554/eLife.28329.003

that, at least under heat stress, cells might relax their ability to confine DNA circles in the mother cell.

## Changes in anaphase duration and nuclear morphology do not explain circle propagation upon heat stress

Three parameters affect the retention of DNA circles in the yeast mother nucleus. First, increased anaphase duration leaves more time to circles to diffuse from the mother into the bud and therefore promotes their propagation (*Gehlen et al., 2011*). Second, a failure to efficiently narrow down the median constriction of the dividing nucleus leaves opportunity to nucleoplasmic, but not membrane attached, material to exchange between mother and bud parts of the nucleus (*Gehlen et al., 2011*; *Boettcher et al., 2012*). Third, the presence of a diffusion barrier in the outer nuclear membrane restricts the movement of circles through the bud neck, provided that they are attached to nuclear

pore complexes (NPCs, *Shcheprova et al., 2008*; *Denoth-Lippuner et al., 2014*). Thus, we wondered which of these parameters is affected in cells grown under heat stress.

First, we characterized the effect of heat on both anaphase duration and changes in nuclear morphology. In a strain carrying our model DNA circle, we tagged the outer nuclear membrane protein Nsg1 with GFP. Using this marker, we quantified the duration of nuclear division (*Figure 2A*, *Figure 2—figure supplement 1A*) and the morphology of the nuclei (length of longitudinal axis and diameter of their scission constriction) throughout anaphase (*Figure 2—figure supplement 2A*). We defined anaphase as the time window starting with the entry of a nuclear lobe into the bud and finishing with the completion of karyokinesis (the two separate nuclei move slightly toward each other, *Figure 2A*, *Figure 2—figure supplement 1A*). Compared to optimal growth conditions, exposure to heat shortened anaphase duration by about 20% (from 19.7 ± 0.3 to 15.7 ± 0.5 min, *Figure 2B*). The length of the nucleus was increased in the first 100 seconds of anaphase but the scission constriction at the bud neck was unaffected (*Figure 2—figure supplement 2B–D*). If anything, an increased length of the nucleus would reduce, and not increase, the DNA circle propagation to the bud. Thus, at first sight the increased propagation of circles in populations of yeast cells grown under mild heat stress was not due to an overall prolongation of anaphase duration or change in nuclear morphology.

## DNA circles segregate during early anaphase

However, the morphology of the nucleus strongly changes between early and late anaphase. To address whether the duration of one of the stages of anaphases was particularly affected and how this could influence DNA circle exchange between mother and daughter cell, we studied when exactly during anaphase DNA circles are exchanged between the mother and daughter part of the nucleus. Based on this, we then examined whether heat stress prolonged that particular stage. We took advantage of the strain described above and followed now simultaneously nuclear division and the segregation of the DNA circles (*Figure 2—figure supplement 1B*). In movies of wild type cells grown at 30°C in medium containing 2% glucose, we determined the propagation flux of DNA circles through the bud neck (number of passages to the bud per minute) and compared the values obtained during early versus late anaphase. This frequency was six to eight folds higher in early anaphase than in late anaphase (0.03 ± 0.01 and 0.004 ± 0.003 passage per minute, respectively, *Figure 2C*). Thus, DNA circles exchange between mother and bud essentially during early anaphase. An extension of early anaphase duration at the cost of late anaphase could increase the frequency at which circles pass to the bud, while maintaining a constant duration of the total anaphase.

Therefore, using the same movies, we determined the time cells spent in early anaphase. Remarkably, early anaphase was shorter, certainly not longer, in cells grown at 37°C, compared to 30°C (3.4 ± 0.1 versus 3.9 ± 0.14 min, *Figure 2D*). The duration of late anaphase was reduced as well (12.9 ± 0.46 versus 14.5 ± 0.46 min, *Figure 2—figure supplement 1C*). Furthermore, early anaphase lasted the longest upon calorie restriction (5.2 ± 0.36 min, *Figure 2D*), although this condition did not affect the retention of DNA circles (*Figure 1C*). Thus, the increased propagation of the DNA circle to the bud in cells grown at 37°C was not due to an extension of the duration of early anaphase.

## DNA circles remain anchored to NPCs upon heat stress

The high-fidelity retention of DNA circles into the mother cell requires their anchorage to NPCs, in order to subject them to confinement by the diffusion barrier in the nuclear envelope (*Shcheprova et al., 2008*; *Denoth-Lippuner et al., 2014*). Thus, we tested whether DNA circle-NPC interaction was affected in cells grown at 37°C. The co-localization of DNA circles with NPCs was measured as previously described (*Denoth-Lippuner et al., 2014*). Intensity profiles of Nup82 labeled with 3x super folder GFP (Nup82-3x sfGFP) were obtained along the nuclear envelope in equatorial focal sections of the nuclei containing a single mCherry-labeled DNA circle at the rim (*Figure 3A*). Nup82-3x sfGFP intensity profiles from at least 40 cells were aligned relative to the maximum intensity of the DNA circle and averaged. When the circle anchors to the NPC, a local Nup82-3x sfGFP intensity peak correlates with the intensity peak of the DNA circle (*Figure 3B,C*). If the DNA circle-NPC interaction is compromised, for example by the knock-out of the acetyltransferase Gcn5 in the SAGA complex (*Denoth-Lippuner et al., 2014*), then the Nup84 fluorescence is not

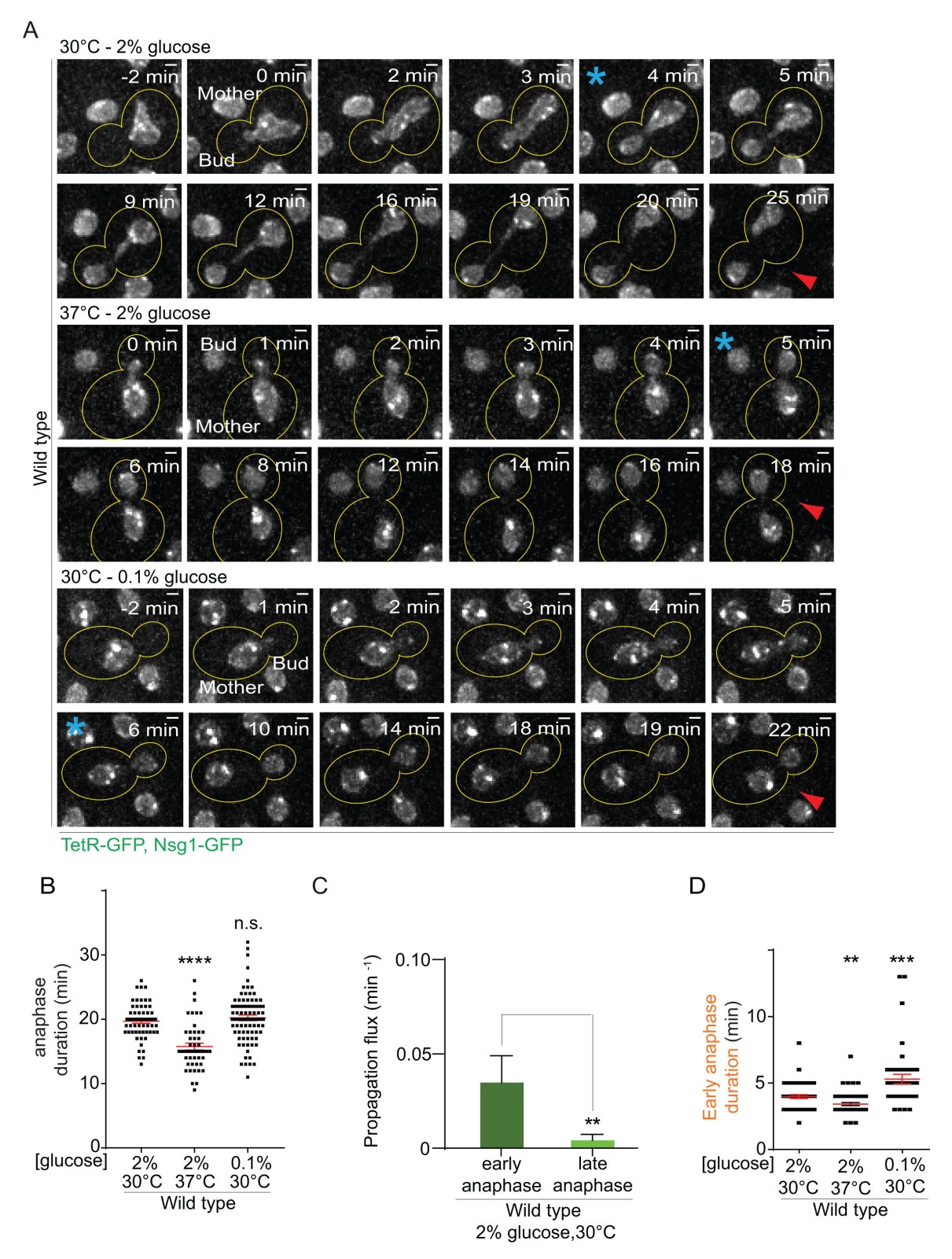

**Figure 2.** Changes in anaphase duration cannot explain DNA circle segregation upon heat stress. (**A**) Dividing nuclei used to measure the duration of anaphase (maximum intensity projection, outline of the cell in yellow, scale bar is 1 μm). Bright dots represent either DNA circles or, if at the opposite sides of the nucleus, the SPBs. For the purpose of the analysis presented here, we measured only the timing of nuclear division without looking at the behavior of the DNA circles. 0 min was set as the beginning of anaphase. The red arrow marks karyokinesis, whereas the blue asterisks mark the end of

*Figure 2 continued on next page*

*Figure 2 continued*

early anaphase (see *Figure 2—figure supplement 1*). B) Quantification of anaphase duration (mean ±SEM of three to four independent experiments with 47 < n < 83 cells and two or three independent clones per condition). Non-parametric Mann-Whitney U test: ****p<0.0001. (C) The propagation flux of individual DNA circles through the bud neck (mean ± SD passage frequency per minute in early and late anaphase of four independent experiments and three independent clones). Number of cells is 51 ≤ n ≤ 104 cells per anaphase stage; Unpaired t-test, **p<0.01. (D) Quantification of early anaphase duration (mean ±SEM of two to three independent experiments per condition), 47 ≤ n ≤ 53 cells per condition. Non-parametric Mann-Whitney U test: **p<0.01.

DOI: https://doi.org/10.7554/eLife.28329.004

The following figure supplements are available for figure 2:

**Figure supplement 1.** Changes in early or late anaphase duration cannot explain DNA circle segregation upon heat stress.
DOI: https://doi.org/10.7554/eLife.28329.005

**Figure supplement 2.** The morphology of the dividing nucleus is not significantly affected by heat stress.
DOI: https://doi.org/10.7554/eLife.28329.006

in phase with the DNA circle fluorescence and the signal correlation is lost (*Figure 3—figure supplement 1*). Interestingly, the correlation between DNA circle and NPC remained intact upon heat shock. Thus, the increased frequency of circle propagation into the daughter upon growth at higher temperature is not due to circles detachment from NPCs.

Together, these data indicated that the increased frequency at which circles propagate to daughter cells in cells grown at 37°C was not due to changes in nuclear morphology, an increase of anaphase duration or circle detachment from NPCs. Therefore, we envisioned the possibility that it might be due to the diffusion barrier, normally present in the outer nuclear membrane, being impaired in cells grown at 37°C.

## Effect of heat stress and calorie restriction on the nuclear diffusion barrier

Thus, we probed the diffusion barrier in the outer nuclear membrane using Fluorescence Loss In Photobleaching (FLIP [*Bolognesi et al., 2016*]) in cells grown at 30°C or 37°C in medium containing 2% or 0.1% glucose. In cells expressing the nucleoporin Nup49 tagged with GFP (Nup49-GFP, nuclear membrane reporter), a small region of the mother part of anaphase nuclei was constantly photo-bleached over time and the fluorescence decay in both mother and daughter nuclear compartments was measured (*Figure 4A,B*). The ratio between the time it took to lose 25% of the signal in the non-bleached compartment (bud) to the time it took to lose 25% of the signal in the bleached compartment (mother) was computed and is defined as the Barrier index (BI, *Shcheprova et al., 2008*, *Figure 4C*). A weaker membrane compartmentalization between mother and bud results in a faster signal decay in the bud, thus a weaker barrier (low BI). These experiments established that barrier strength in the nuclear membrane was reduced by roughly half in cells grown at 37°C compared to those grown at 30°C (BI = 24.6 ± 2.5 vs 41.9 ± 4.1; *Figure 4D*). As a positive control, the barrier index in the *bud6Δ* mutant cells grown at 30°C, which bear strong barrier defects (*Shcheprova et al., 2008*), was similarly decreased (BI = 25.3 ± 3.2). In contrast, and to our surprise, the barrier was significantly strengthened in calorie-restricted cells grown in 0.1% glucose (BI = 68.6 ± 15, *Figure 4D*). The diffusion barrier in the cortical ER was not affected by heat stress and calorie restriction (*Figure 4—figure supplement 1A–C*). Thus, the effect of temperature and calorie restriction targeted specifically the diffusion barrier in the outer nuclear membrane. We conclude that both heat stress and calorie restriction affect the nuclear diffusion barrier, but in opposite manners. These results suggest that the increased propensity with which daughters inherit circles upon growth at 37°C might be due to a reduction in barrier strength.

Furthermore, the measured increase of barrier strength upon calorie restriction (*Figure 4D*) might explain why these cells, while spending more time in early anaphase (*Figure 2D*), do not segregate circles more frequently to their daughters (*Figure 1C*). We expect that during early anaphase, the propagation flux of individual DNA circles through the bud neck (number of passages to the bud per unit of time, independent of anaphase duration) is increased at 37°C and decreased in calorie restricted cells, assuming that the DNA circles retention depends on the diffusion barrier. Indeed, the propagation flux in early anaphase was increased two folds in cells grown at 37°C compared to 30°C (0.06 ± 0.004 versus 0.03 ± 0.01 passage per minute; *Figure 4E*) and decreased three folds in

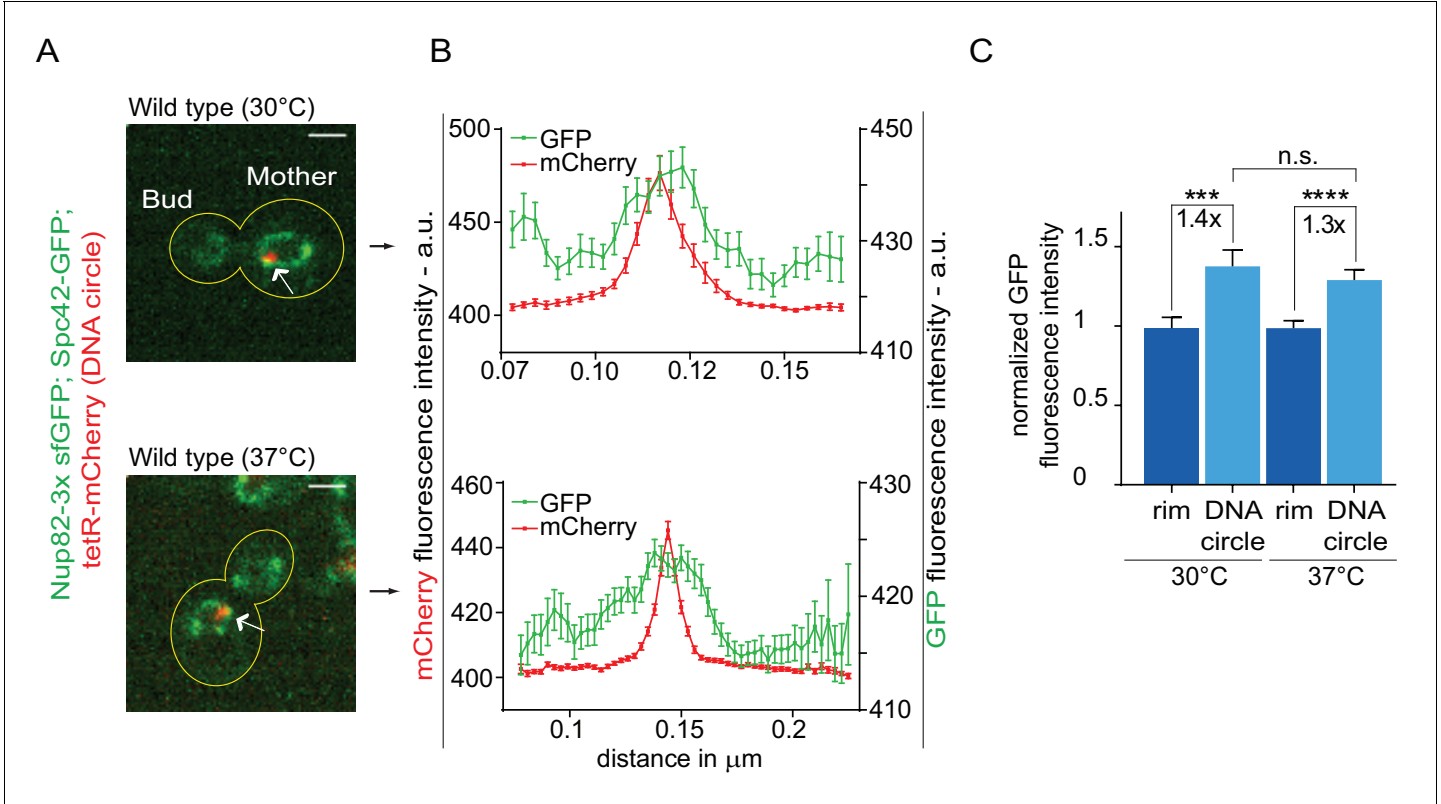

**Figure 3.** Heat stress does not affect the attachment of DNA circles to the NPCs. (**A**) Representative anaphase nuclei (one focal plane, outline of the cell in yellow, scale bar is 1 μm). The DNA circle is in red (tetR-mCherry, arrow) and the NPCs in green (Nup82-3x sfGFP). (**B**) Fluorescence intensity profiles for Nup82-3x sfGFP were aligned with respect to the maximum intensity of the DNA circle tetR-mCherry intensity peak (Mean ± SEM). **C**) The mean Nup82-GFP fluorescence intensity at the entire rim of the nuclear envelope (normalized to 1) or locally at the DNA circle, from 41 ≤ n ≤ 105 cells per condition (mean ± SEM). The fold changes of Nup82-GFP intensity at the DNA circle versus the rest of the nuclear envelope are indicated. Four independent pooled experiments per condition. Unpaired t-test, ***p<0.001.

DOI: https://doi.org/10.7554/eLife.28329.007

The following figure supplement is available for figure 3:

**Figure supplement 1.** DNA circle are detached from NPCs in SAGA-mutant strain.

DOI: https://doi.org/10.7554/eLife.28329.008

calorie restricted cells (0.01 ± 0.01 passage per minute). The decreased propagation flux in combination with a longer anaphase duration (*Figure 2D*) yielded an unaltered DNA circle propagation frequency in calorie restricted cells (*Figure 1C*). The increased propagation flux in heat-stressed cells is comparable to that in the *bud6Δ* mutant cells (0.10 ± 0.02 passage per minute, *Figure 4E*). Thus, heat stress affected the permeability of the bud neck for DNA circles. We conclude that the permeability of the diffusion barrier emerged as the tightest and most direct determinant of circle retention in the mother cell.

## Constitutive activation of the Pkc1 stress response kinase recapitulates the effect of heat stress on the nuclear diffusion barrier

The observation that heat stress had a specific effect on the nuclear barrier and not the cortical ER barrier, hinted toward a regulated process instead of a general effect of temperature for example on membrane fluidity. To address this possibility, we investigated whether stress response pathways regulate barrier strength. We particularly focused on the possible role of the cell wall integrity pathway (CWI), which is activated upon heat stress. At the top of this pathway, the Pkc1 kinase responds to plasma-membrane and cell wall stress and activates a MAP-kinase cascade to promote cell wall remodeling and repair (*Levin, 2005*, *Figure 5A*). Thus, we asked whether constitutively activating this pathway affected the strength of the barrier in the nuclear membrane.

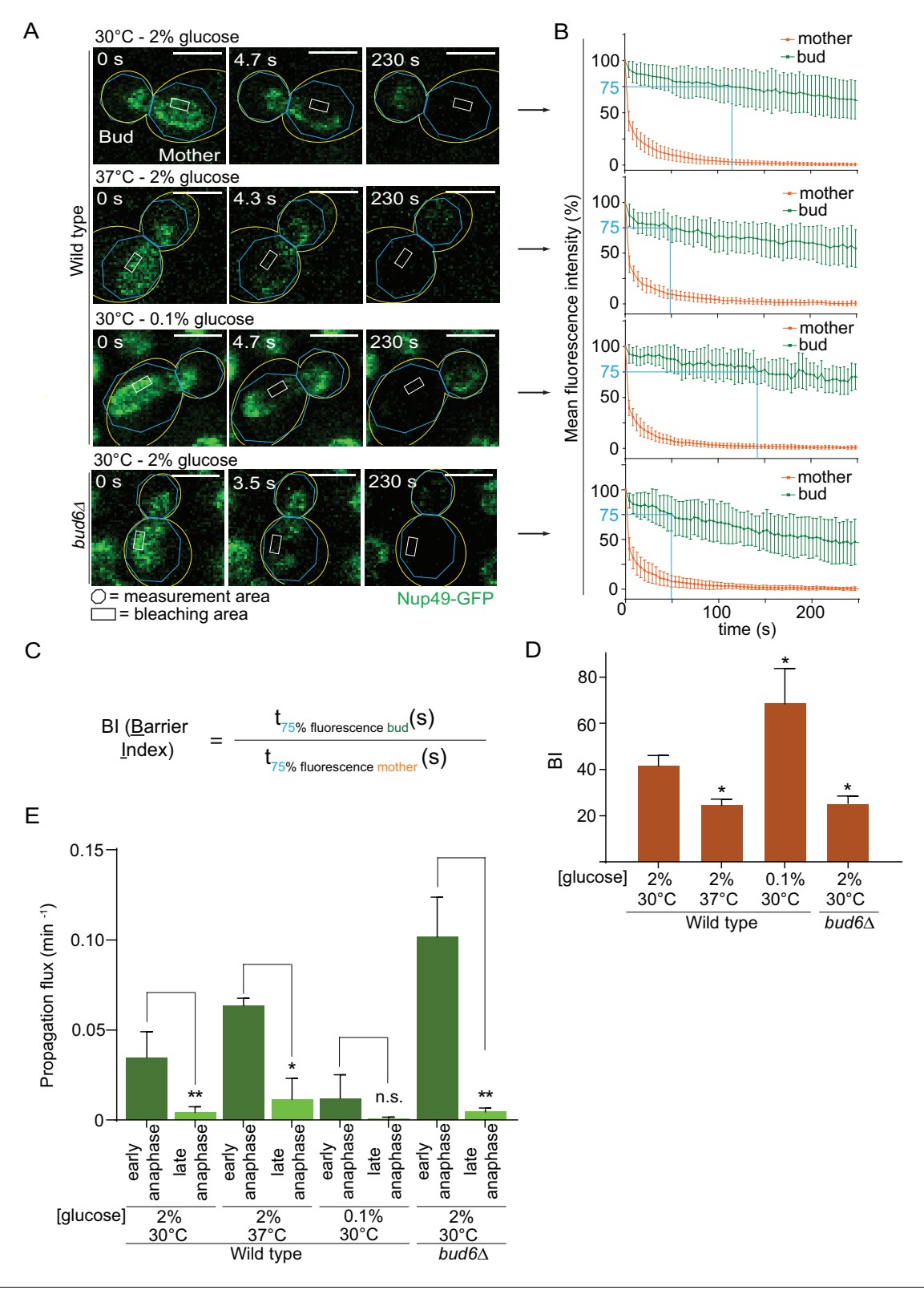

**Figure 4.** Heat stress and calorie restriction affect the nuclear diffusion barrier in opposite manners. (**A**) Representative dividing nuclei (outline of the cell in yellow, scale bar is 3 μm). Areas of constant bleaching and fluorescence measurement are indicated. (**B**) Mean fluorescence intensity over time (±SD, 18 ≤ n ≤ 55 cells per condition). The blue lines illustrates the 75% threshold used to define the barrier index (see method for details). (**C**) Definition of the BI. $t_{75\% \text{ fluorescence}}$ is the time required for the fluorescence to decrease to 75% of its initial value. The BI is defined as the ratio of $t_{75\%}$

*Figure 4 continued on next page*

Figure 4 continued

fluorescence in the non-bleached (bud) compartment to that of the bleached (mother) compartment. (D) BI quantification of measurements in the nuclear envelope (±SEM, unpaired t-test, *p<0.05). (E) The propagation flux of individual DNA circles through the bud neck (mean ±SD passage frequency per minute in early and late anaphase of two to four independent experiments and three independent clones per condition). Number of cells as follows: 37°C (2% glucose), 51 ≤ n ≤ 104 cells per anaphase stage; 30°C (0.1% glucose), 35 ≤ n ≤ 39 cells per anaphase stage; bud6Δ mutant cells, 21 ≤ n ≤ 27 cells per anaphase stage. Unpaired t-test, *p<0.05. Note that the data for 30°C (2% glucose) is a duplication from *Figure 2*, for comparison purposes.
DOI: https://doi.org/10.7554/eLife.28329.009
The following figure supplement is available for figure 4:

**Figure supplement 1.** Heat stress and calorie restriction do not affect the cortical ER diffusion barrier.
DOI: https://doi.org/10.7554/eLife.28329.010

To this end, we introduced the constitutively active allele of the *PKC1* gene, *PKC1-R398P* (*Nonaka et al., 1995*) in our wild-type strain expressing our reporter nucleoporin (Nup49-GFP) and applied FLIP to determine its effect on barrier strength in cells grown at 30°C. Supporting the idea that Pkc1 regulates the nuclear barrier, cells expressing this constitutively activated form of Pkc1 showed a much weaker barrier compared to wild type cells (BI 19.1 ± 2 versus 41.9 ± 4.1, *Figure 5B*, *Figure 5—figure supplement 1*) and a significant reduction in DNA circle retention (*Figure 5C*). Strikingly, this effect did not require the MAP-kinase cascade downstream of Pkc1. Constitutive activation of the MAP kinase kinase kinase Bck1, using the *BCK1-20* allele, did not change barrier strength (BI = 38.4 ± 7.1, *Figure 5B*), compared to wild-type cells. Furthermore, inactivating the MAP kinase Slt2, which acts most downstream in the CWI pathway (*Figure 5A*), did not revert the effect of the *PKC1-R398P* mutation (BI = 17.9 ± 2.1 in the *PKC1-R398P slt2Δ* double mutant cells, *Figure 5B*, *Figure 5—figure supplement 1*). Finally, instead of promoting barrier strength, the *slt2Δ* mutation by itself tends to slightly weakening the barrier strength (BI = 28.5 ± 4.2), fitting with the cell wall defects observed in these cells. Thus, these data indicate that activation of the Pkc1 kinase inhibits the diffusion barriers in the nuclear envelope and that this effect depends on a distinct signaling branch than the MAP-kinase cascade. These data support the conclusion that the weakening of the diffusion barrier in heat-treated cells corresponds to a regulatory response of the cells and not a direct effect of temperature on barrier structure or function.

## PKA and Tor1 inhibit the nuclear diffusion barrier

Among others, the two kinases PKA and Tor1 also contribute to the cellular responses to stress (*Causton et al., 2001*; *Castells-Roca et al., 2011*; *Loewith and Hall, 2011*; *Pautasso and Rossi, 2014*). They also promote cell growth in response to nutrients availability and are down-regulated in response to calorie restriction (*Thevelein and de Winde, 1999*; *Loewith and Hall, 2011*), *Figure 5A*). Based on this and the fact that the cells increase barrier strength upon calorie restriction (*Figure 4D*), we tested whether PKA and Tor1 inhibited the nuclear diffusion barrier. Constitutive activation of PKA through deleting the *BCY1* gene that encodes its inhibitory subunit (*Toda et al., 1987*) significantly reduced barrier strength compared to wild type cells grown in the same conditions (BI$_n$ 19.9 ± 1 versus 41.9 ± 4.1, *Figure 5D*, *Figure 5—figure supplements 1–2*). Likewise, expression of the constitutive active allele of Tor1, *TOR1-A1957V* (*Reinke et al., 2006*) had the same effect (BI$_n$ = 20.75 ± 1.9, *Figure 5D*, *Figure 5—figure supplements 1–2*). In reverse, partial inhibition of Tor1 by addition of rapamycin (200 ng/ml for 16–18 h), a TORC1-specific inhibitor, to the growth medium (*Brunn et al., 1996*; *Loewith and Hall, 2011*) increased barrier strenght, even more than lowering glucose concentration (BI$_n$ up to 107 ± 24 and 68.6 ± 15, respectively *Figure 5D*, *Figure 5—figure supplement 1*). These data indicate that the Tor1 and PKA kinases act in pathways inhibiting the nuclear diffusion barrier in rich medium and perhaps in response to stress, similarly to Pkc1.

## Tor1 and PKA act downstream of Pkc1 in the response to heat stress

To investigate whether Tor1 and PKA activity contribute to barrier weakening in response to heat stress, we took advantage of calorie restriction inhibiting both PKA and Tor1 (*Steinkraus et al., 2008*; *Fontana et al., 2010*). Thus, we tested whether cells grown in low glucose, that is, with low PKA and Tor1 activity, were still able to repress barrier function in response to heat stress. Whereas cells grown at 37°C in 2% glucose decreased the barrier strength compared to wild-type cells grown

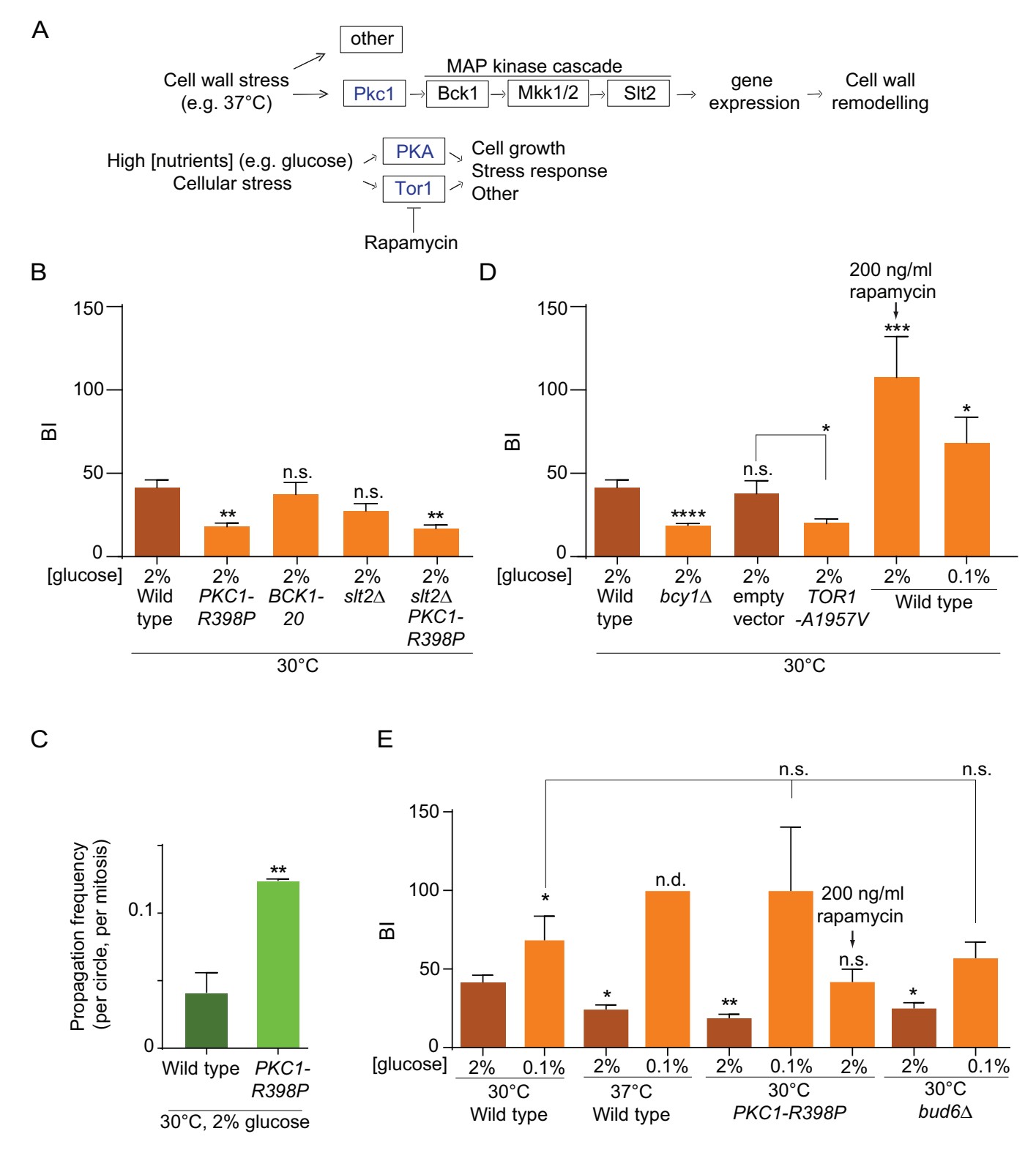

**Figure 5.** Effect of PKA, Pkc1 and Tor1 stress response kinases on the nuclear diffusion barrier. (A) Simplified scheme of the Pkc1, Tor1 and PKA stress response pathways. (B) Quantification of BI in nuclear envelope upon perturbations in cell wall integrity pathway. PKC1-R398P and BCK1-20 are constitutive active alleles of Pkc1 and Bck1. (C) Propagation frequency: mean ±SEM of three experiments with $27 \leq n \leq 51$ cells per experiment; unpaired t-test, **p<0.01. (D) Quantification of BI in the nuclear envelope upon perturbations in PKA and Tor1 pathway (E) and after restoration of the
*Figure 5 continued on next page*

*Figure 5 continued*

diffusion barrier. TOR1-A1957V is a constitutive active allele of Tor1. For (**B**), (**D**), (**E**): BI ± SEM, 16 ≤ n ≤ 56 cells per condition, unpaired t-test, *p<0.05. For comparison purposes: the BI of wild-type cells (37°C and 0.1% glucose) was arbitrarily set to 100 (n.d., see main text). The BI values of wild type cells (2% and 0.1% glucose, 30°C and 37°C) and of *bud6Δ* (2% glucose, 30°C) in *Figure 5B,D and E* are the same as in *Figure 4D*, the BI of PKC1-R398P mutant cells (2% glucose, 30°C) in *Figure 5E* is the same as in *Figure 5B*.

DOI: https://doi.org/10.7554/eLife.28329.011

The following figure supplements are available for figure 5:

**Figure supplement 1.** Effect of the CWI and Tor1 pathways on the nuclear diffusion barrier.

DOI: https://doi.org/10.7554/eLife.28329.012

**Figure supplement 2.** Effect of PKA and Tor1 on the nuclear diffusion barrier in response to heat stress.

DOI: https://doi.org/10.7554/eLife.28329.013

in the same medium at 30°C (BI = 24.6 ± 2.5, versus 41.9 ± 4.1, *Figure 5E*), calorie restriction abrogated this effect. In fact, cells grown at 37°C in the calorie restricting medium (0.1% glucose) formed a stronger barrier (*Figure 5E*, *Figure 5—figure supplements 1–2*). We could not measure the BI in these cells, because in average the fluorescence failed to decay significantly in the bud. Calorie restriction had a similar effect on the *PKC1-R398P* mutant cells grown at 30°C. Here again, calorie restriction did not simply restore the barrier of the mutant cells, but enhanced it compared to growth under optimal conditions (BI = 100 ± 40 in 0.1% glucose versus 17.9 ± 2.1 in 2% glucose, *Figure 5E*, *Figure 5—figure supplements 1–2*). Interestingly, *PKC1-R398P* mutant cells grown at 30°C (in 2% glucose) and treated with rapamycin (200 ng/ml), restored the barrier strength to levels similar to those observed in wild-type cells (BI = 42.2 ± 7.7, *Figure 5E*, *Figure 5—figure supplements 1–2*). These data suggest that 37°C and Pkc1 regulate barrier strength upstream of Tor1 and PKA, and that Tor1 and/or PKA activity is required in order to repress barrier function in response to heat stress. Strikingly, calorie restriction also restored the barrier in the *bud6Δ* mutant cells compared to growth in 2% glucose (BI = 57 ± 9.9 versus 25.3 ± 3.2, respectively; *Figure 5E*, *Figure 5—figure supplements 1–2*). Thus, the barrier defect of *bud6Δ* mutant cells is at least in part suppressed by inhibiting the Tor1 and PKA pathways. Collectively, our data indicate that the strength of the nuclear diffusion barrier is a regulated trait under the control of Tor1 and PKA.

## Restoration of the diffusion barrier during heat stress rescues DNA circle confinement in the mother cell

We next sought to directly test whether the increased DNA circle propagation frequency observed upon heat stress is indeed due to their weaker diffusion barrier in the nuclear envelope. We reasoned that if it were the case, strengthening the barrier by calorie restriction (as in *Figure 4D*) in cells grown at 37°C should reduce the propagation of DNA circles to the bud, normally observed upon heat stress. Thus, we examined the propagation frequency of the DNA circle in cells grown at 37°C in medium containing 0.1% glucose. At 30°C (0.1% glucose) the propagation frequency was 0.04 ± 0.005, similar to the 2% glucose condition (0.04 ± 0.008, *Figure 6A,B*). Interestingly, in cells grown at 37°C calorie restriction restored the propagation frequency to the levels normally observed at 30°C (0.06 ± 0.01, compared to 0.16 ± 0.02 in 2% glucose, *Figure 6A,B*). In line with the reinforcing effect of calorie restriction on the barrier, *bud6Δ* mutant cells grown at 30°C in 0.1% glucose containing medium also retrieved their ability to confine the DNA circle into the mother cell (propagation frequency = 0.04 ± 0.007, compared to 0.12 ± 0.01 in 2%, *Figure 6A,B*). We conclude that restoring barrier strength under heat stress conditions rescues the retention of DNA circles in the mother cell, indicating that the increased propagation of DNA circles at elevated temperatures (*Figure 1C*) is indeed caused by a down regulation of the diffusion barrier.

## Barrier relaxation promotes lifespan extension upon heat stress

Collectively, our data indicate that heat stress, which causes lifespan extension (*Shama et al., 1998a*; *Swieciło et al., 2000*), relaxes the confinement of DNA circles in the yeast mother cell through weakening of the diffusion barrier. Thus, our data predicts a reduction in DNA circle content when the aged mother cells were grown at 37°C instead of 30°C. Therefore, we monitored by Southern blotting the levels of rDNA circles in aged cells, incubated at both 30°C and 37°C, as described

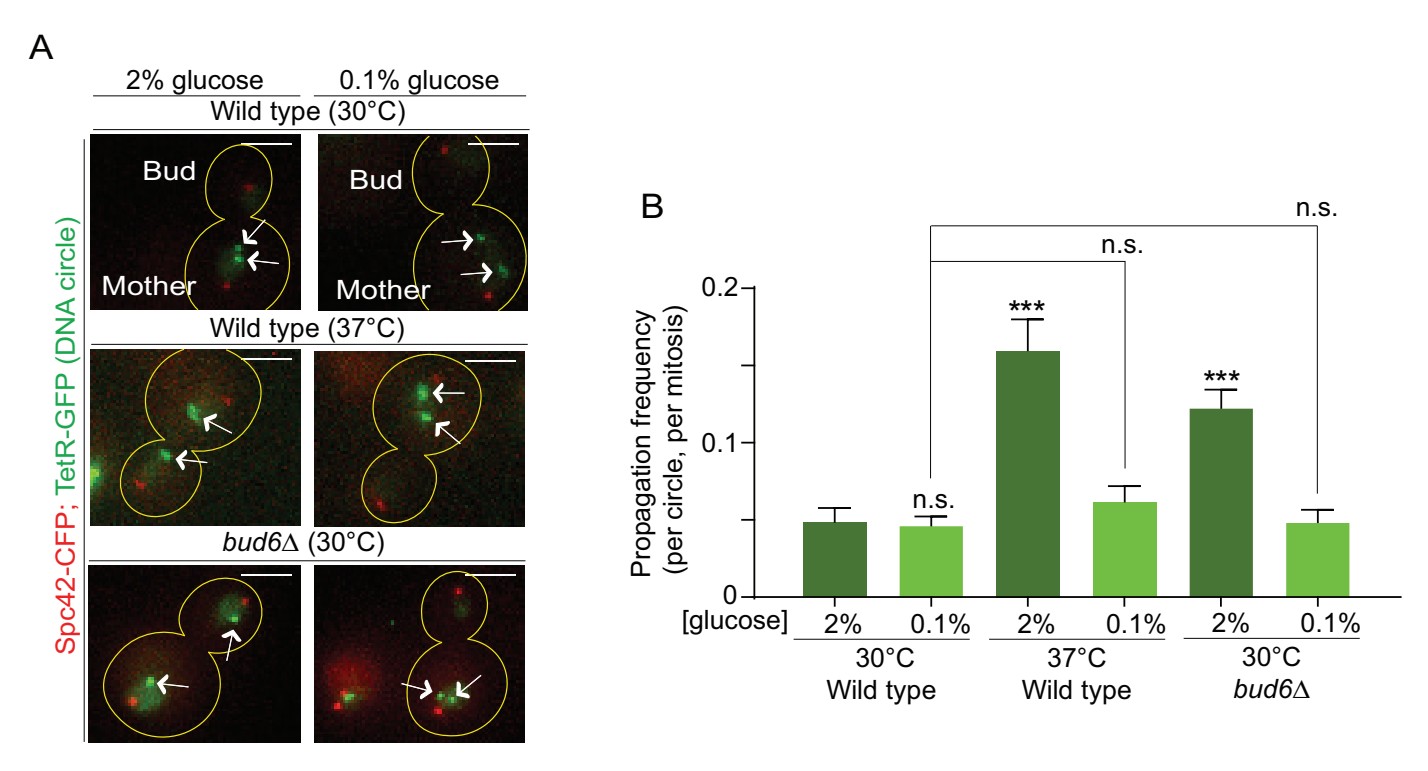

**Figure 6.** Restoring the nuclear barrier during heat stress rescues DNA circles confinement in the mother cell. (**A**) Examples of anaphase cells (max intensity projection, outline of the cell in yellow, scale bar is 3 μm). The arrows indicate DNA circles. The SPBs appear in red due to image processing. (**B**) Propagation frequency, mean ± SEM of several independent experiments per condition as follows: $N = 6$ (3 from **Figure 1**C + 3 new) for both wild type ($52 \leq n \leq 125$ per experiment) and $bud6\Delta$ mutant cells ($61 \leq n \leq 122$ per experiment) at 30°C (2% glucose); $N = 6$ for both wild type ($31 \leq n \leq 74$ per experiment) and $bud6\Delta$ mutant cells ($32 \leq n \leq 156$ per experiment) at 30°C (0.1% glucose); $N = 5$ for wild-type cells at 37°C (0.1% glucose, $72 \leq n \leq 184$ per experiment); $N = 4$ (3 from **Figure 1**C + 1 new) for wild-type cells at 37°C (2% glucose, $74 \leq n \leq 98$ per experiment). Unpaired t-test, ***$p < 0.001$. For comparison purposes the data of wild-type cells at 30°C (0.1% glucose) is the same as in **Figure 1**.
DOI: https://doi.org/10.7554/eLife.28329.014

(**Denoth-Lippuner et al., 2014**). We enriched for aged yeast mother cells (15% of aged cells, 10 000 fold enrichment; average age of the aged cells: 16 generations old), using the mother enrichment program (**Lindstrom and Gottschling, 2009**). We extracted total DNA from young and aged cell populations. Similar amounts of DNA (300 ng) were used, from young cell populations and aged cell populations with same fraction of aged cells. A [32]P-labeled probe specific for the rDNA locus was used to detect the ERCs. While no DNA circles were observed in young cells, a substantial amount could be seen in the old ones. And indeed, cells cultured at 37°C showed 3.3-fold less ERCs than cells cultured at 30°C (**Figure 7A**). We concluded that cells grown at 37°C indeed decrease the amount of ERCs that they accumulate, consistent with DNA circle retention being relaxed.

Thus, our data opened the possibility that barrier weakening is a mechanism through which cells regulate the accumulation of DNA circle with age, and hence, their longevity in response to heat stress. We reasoned that if it where the case then restoring the diffusion barrier in heat stressed cells, using calorie restriction (see **Figure 4D**), should reduce their longevity to the level of unstressed cells. Thus, we compared the longevity of heat stressed and calorie restricted cells, and cells subjected to both conditions at the same time, using standard micro-dissection technics and pedigree analysis.

Whereas wild-type cells grown under optimal conditions showed a median lifespan of 23 generations, calorie restriction increased their median lifespan to 30 generations at 30°C (**Figure 7B**), as reported (**Fontana et al., 2010**). As published (**Shama et al., 1998a**; **Shama et al., 1998b**; **Swieciło et al., 2000**), growth at 37°C increased the longevity to a median lifespan of 32.5 generations (**Figure 7B**). Calorie restriction of cells grown at 37°C shortened their lifespan back to

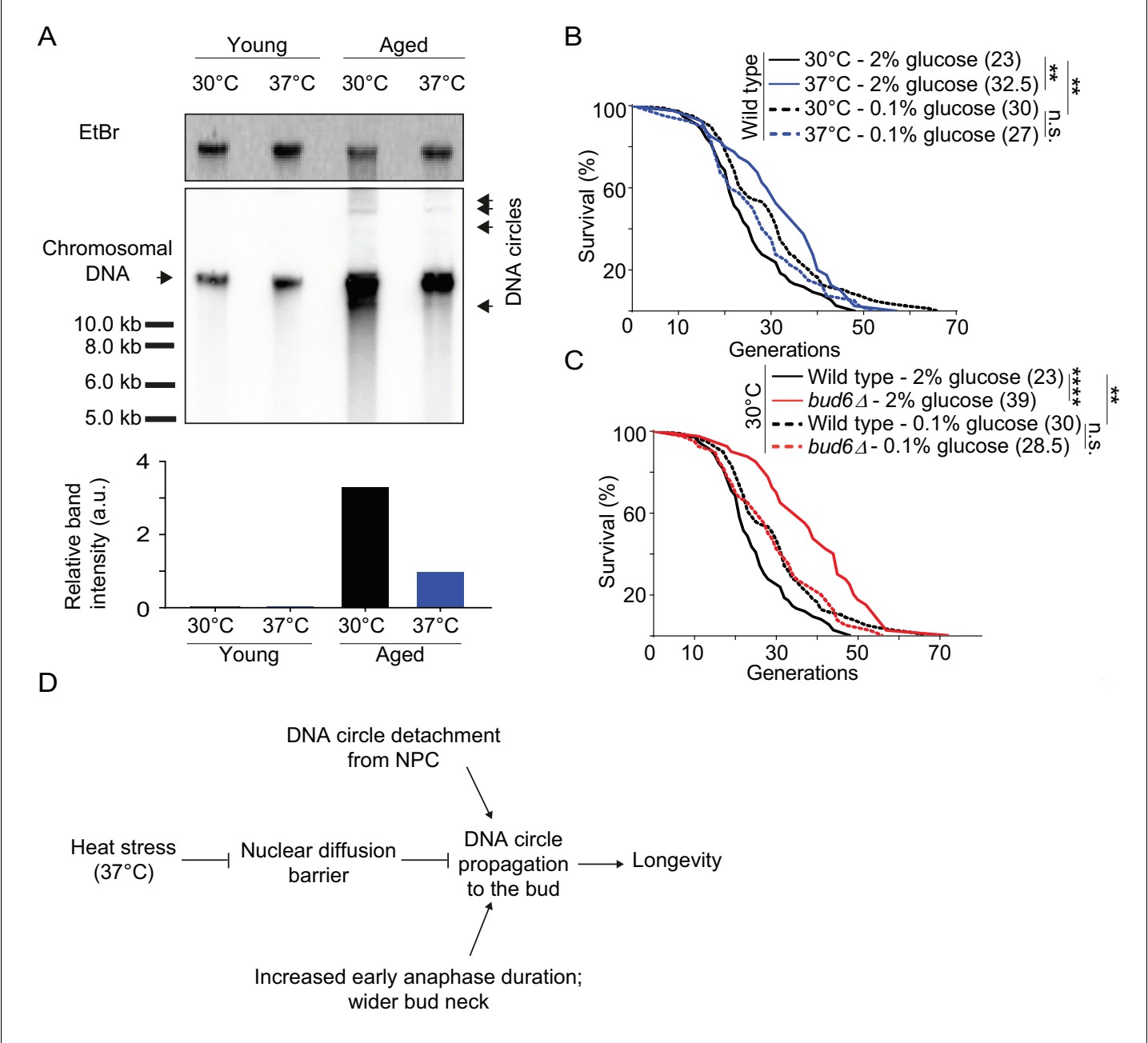

**Figure 7.** Impaired nuclear barrier strength correlates with an increased lifespan. (**A**) Detection of DNA circles by Southern blotting in young and aged cells grown at 30°C and 37°C (2% glucose). Young cells were 0–1 generation and aged cells were 16 generations old. The quantification of the relative band intensity is based on four bands of concatenated DNA circles. Band sizes of a DNA marker are indicated. (**B–C**) Lifespan analysis. Median lifespan (in brackets) of $40 \leq n \leq 79$ cells per condition. two independent pooled experiments per condition. For comparison purposes: the lifespan curves of wild type cells (2% and 0.1% glucose, 30°C) in *Figure 7A* and *Figure 7B* are the same. Log Rank (Mantel-Cox) test, **p<0.01. (**D**) Simplified: heat stress specifically weakens the nuclear diffusion barrier, thus fostering a more symmetric segregation of DNA circles to the daughter and accordingly promoting longevity.

DOI: https://doi.org/10.7554/eLife.28329.015

27 generations (*Figure 7B*). While the *bud6Δ* mutant cells grown at 30°C in rich medium showed an increased lifespan (median lifespan = 39), as reported (*Shcheprova et al., 2008*, *Figure 7C*), calorie restriction shortened their longevity back to a value close to that of wild type cells under the same conditions (median lifespan = 28.5; *Figure 7C*). Thus, aging of the bud6Δ mutant and heat stressed

cells was restored upon strengthening the diffusion barrier back to wild type or higher levels. This is consistent with the idea that heat stress increases the longevity of the mother cell at least in part through the inhibition of the diffusion barrier in the nuclear envelope and consequently through releasing aging factors, such as DNA circles, to their progeny. Collectively, our data suggest that the redistribution of age load, such as DNA circles, between mother and daughter is a mechanism for modulating the longevity of the cell in response to stresses such as heat (*Figure 7D*).

## Discussion

### Yeast cells modulate their longevity via diffusion barrier strength regulation under heat stress

The efficient retention of DNA circles in the mother cell during mitosis is a key determinant of the replicative lifespan of *S. cerevisiae* cells. How cells achieve this retention, has been a topic of debate (*Gehlen et al., 2011*; *Khmelinskii et al., 2011*; *Ouellet and Barral, 2012*; *Denoth-Lippuner et al., 2014*). Our data provide evidence confirming the nuclear diffusion barrier as being a key parameter contributing to DNA circle retention and reveals that the strength of the barrier is regulated. First, we observed that the barrier loses permeability in cells grown at 37°C, increasing propagation of DNA circles to the bud. Second, the barrier permeability is restored in these cells upon calorie restriction which leads to full restoration of circle retention in the mother cell. These results are consistent with the observation that the attachment of DNA circles to NPCs is required for their efficient retention in the mother cell, in a barrier-dependent manner (*Shcheprova et al., 2008*; *Denoth-Lippuner et al., 2014*). Importantly, heat stress did not impair NPC-DNA circle association, indicating that it does not affect DNA circle propagation via their detachment from the nuclear envelope.

Furthermore, our data indicate that regulation of barrier permeability under heat stress results in the redistribution of the age load during mitosis, resulting in a longer life span for the mother cell. In other words, the yeast cell is able to modulate longevity by regulating the diffusion barrier. We observed a two-fold reduction of barrier strength during heat stress (*Figure 4D*) and a roughly four-fold increase of DNA circle propagation to the daughter cells at 37°C (*Figure 1C*). Effectively, this represents a decrease of circle retention from 96% to 83% for what concerns the model DNA circle. Is such an apparently mild decrease sufficient to explain the lifespan increase that we observe (*Figure 7B*)? Mathematical modeling studies predicted that a retention probability above 99% is required to simulate the aging curves experimentally established for wild type cells (*Gillespie et al., 2004*). This is in part due to the fact that the replication origin on the rDNA circles is activated only in about 60% of the cell cycles (*Gillespie et al., 2004*), such that the loss of ERCs to the daughter cells is inefficiently compensated by replication of the DNA circles. Accordingly, previous studies confirmed that a slight reduction of this retention efficiency has a strong effect on the longevity of the cell (*Shcheprova et al., 2008*) reviewed in *Denoth Lippuner et al. (2014)*. This fits with the observation here that ERC levels decrease with roughly 3.5-fold in 16 generations old mother cells grown at 37°C compared to 30°C, despite the apparently high retention efficiency of 83% (*Figure 7A*).

### How is the diffusion barrier regulated?

Our study establishes that the modulation of barrier strength is not merely a direct consequence of nutrients limitation or temperature increase on barrier formation or stability, but a process elicited by specific regulatory pathways in response to environmental disturbances. The effect of heat stress on barrier permeability could be mimicked by activation of Pkc1 and prevented by calorie restriction. Remarkably, inhibition of both PKA and Tor1 (via calorie restriction) in cells either grown at 37°C or expressing a constitutively active form of Pkc1 prevented the weakening of nuclear barrier that is normally observed under these conditions. These data suggest that Pkc1 might modulate barrier strength through Tor1 and PKA. To our knowledge, this is first evidence that Pkc1 might regulate TORC1 and PKA, revealing an intriguing link between these pathways that now needs to be substantiated.

# An 'age-sharing' mechanism underlies the longevity effect of heat stress

Besides *S. cerevisiae* (*Shama et al., 1998a*; *Shama et al., 1998b*; *Swieciło et al., 2000*), heat stress promotes longevity in several other species, ranging from flies (*Smith, 1958*; *Lithgow et al., 1995*; *Khazaeli et al., 1997*; *Le Bourg et al., 2001*) to worms (*Butov et al., 2001*; *Cypser and Johnson, 2002*; *Gems and Partridge, 2008*; *Rodriguez et al., 2012*) and rats (*Holloszy and Smith, 1986*). Heat stress is also a protein denaturant agent. Prokaryotic and eukaryotic species exposed to elevated temperatures induce the expression of heat shock proteins (HSPs) that are well conserved among species and help the cell maintain proteostasis (*Lindquist and Craig, 1988*; *Morano et al., 2012*). In budding yeast, several HSPs are induced upon heat stress (*Boy-Marcotte et al., 1999*) and are implicated in refolding heat-denatured proteins, preventing their aggregation or, if severely damaged, targeting them for degradation (*Hilt and Wolf, 2006*; *Craig et al., 1994*; *Parsell et al., 1994*; *Glover and Lindquist, 1998*). Accumulation of damaged yeast (*Aguilaniu et al., 2003*; *Erjavec et al., 2007*; *Hill et al., 2014*; *Saarikangas and Barral, 2015*). Additionally, the lifespan extension observed in *S. cerevisiae* upon heat stress, requires the heat shock protein Hsp104 (*Shama et al., 1998a*; *Shama et al., 1998b*). HSPs expression decreases with ageing in mammals, flies and worms (*Kurapati et al., 2000*; *Wyttenbach et al., 2002*; *Hsu et al., 2003*) and positively correlates with maximum longevity in mammals and birds (*Salway et al., 2011*). Heat stress was also proposed to induce the oxidative stress response, thus protecting cells against Reactive Oxygen Species (ROS), also associated with ageing in *S. cerevisiae* (*Morano et al., 2012*). These observations have suggested that heat stress promotes longevity via induction of "repair" and/or a "clearance" mechanisms that refold denatured molecules, prevent their aggregation and target damaged/toxic species for degradation (*Estruch, 2000*; *Verbeke et al., 2001*; *Calderwood et al., 2009*).

In striking contrast, we show here that a potent mechanism to promote longevity is the opening of the diffusion barrier in the nuclear envelope and the 'dumping' of at least some of the mother's burden onto their daughters. Barrier weakening is clearly not the mechanism promoting longevity in response to calorie restriction, establishing that it is not the only mechanism through which cells can modulate the effects of age. Most likely, the 'repair', 'clearance' and 'sharing/dumping' mechanisms co-exist and their relative impact might vary between conditions. In any case, the fact that rescuing DNA circle confinement largely abolishes the effect of heat stress on lifespan extension indicates that at least under that condition 'dumping' age load onto the population is a prominent mechanism of longevity regulation. Here, we have focused on the characterization of a single type of aging factor. However, the fact that other factors, such as deposit precursors (*Saarikangas et al., 2017*) also depend on the diffusion barrier for their confinement in the mother cell suggest that the regulation we have uncovered might affect more than only DNA circles. Furthermore, other mechanisms might help redistribute aging factors. Our study is probably only scratching the top of a larger panel of possibilities. Indeed, the nature of the aging factors being redistributed to the progeny could depend on the nature of the physiological signals that the cell faces. In addition, the fact that calorie restriction extends the replicative lifespan of yeast cells without affecting circle confinement indicates that barrier opening is not the only mechanism of longevity modulation.

The idea that stress does not lead to higher fidelity but rather less is actually quite satisfying. Indeed, the widespread idea that yeast cells generate less damage or eliminate them better when they are under stress is somehow paradoxical. It seems somewhat counterintuitive that cells would do better when they are stressed than when they are not, although we agree that there might be selection advantages to such a situation. In any case, our data open the question of why do cells activate a pathway for living longer and aging slower in response to stress, and this potentially at the cost of their progeny. Several lines of thoughts merit attention here. One possibility is that dispersing the pre-existing damage through a large population of cells allows the individual cells to better survive. Here, the gain in longevity would merely be a contingent effect. However, another line of thoughts could be that older yeast mother cells have acquired with age something beneficial, helping them to better cope with stress. Sharing this with their progeny might be important for the survival of the population.

Remarkably, middle-age mother cells do cope better with heat than young ones (reviewed in *Denoth Lippuner et al., 2014*). If the yeast mother cells carry some components that provide them

with a selective advantage under stress, then opening the barrier might have two advantages at once: it might not only allow them to share this advantage with their daughters, but by allowing them to dump damage, it could also help keeping these experienced mothers alive longer. Both effects would be to the benefit of the survival of the genotype. In favor of such a model, it has been recently shown that around 30% of the genome can recombine out of the chromosomes and form DNA circles (*Møller et al., 2015*). In some cases, these circles carry genes likely to promote survival in response to specific stresses, such as cupper intoxication and salt stress (*Møller et al., 2015*). Relaxing barrier permeability would still allow circle accumulation in the mother, but at the same time increase the number of daughters that inherit at least few circles and can accumulate them as well.

Another supportive scenario relates to the observation that protein aggregates that are retained in the mother cell can provide adaptive advantages. This is for example the case when the Whi3 mnemon aggregates in response to futile pheromone exposure and mediates adaptation of the cell to the lack of a partner (*Caudron and Barral, 2013*). Provided that mnemon asymmetry also depends on ER compartmentalization, opening the barrier would allow the mother to share adaptive circles and mnemons with her progeny and at the same time to soften of the pro-aging effect that these factors have on her.

Whether the 'dumping/sharing' mechanism is conserved beyond yeast is unknown. However, it has recently become clear that diffusion barriers are conserved in metazoans (*Moore et al., 2015*; *Lee et al., 2016*). Furthermore, at least the diffusion barrier of neural stem cells changes strength during development and with age (*Moore et al., 2015*). Thus, barrier regulation might be a recurring way to modulate the distribution of age and fate determinants between sister cells in many organisms, and hence a widely used mechanism for modulating the longevity of diverse cell types.

## Materials and methods

### Reagents, strains, and growth conditions

All used strains were constructed according to standard genetic techniques (*Janke et al., 2004*) and are isogenic to S288c (*Winzeler et al., 1999*, *Table 1*). The strains carrying Nsg1-GFP, Nup49-GFP, and Nup82-3GFP are from the genome-wide GFP collection (*Huh et al., 2003*). The plasmids expressing the PKC1-R398P, *Bck1-20* and TOR1-A1957V alleles were already described (*Nonaka et al., 1995*; *Helliwell et al., 1998*; *Reinke et al., 2006*) and the first two were a kind gift of Michael Hall. Concerning the experiments testing the effect of Pkc1, Bck1 and Tor1 on the diffusion barrier (*Figure 5*): the plasmids expressing PKC1-R398P, *Bck1-20*, and the empty vector (pRS315) were introduced as centromeric plasmids in the strain carrying Nup49-GFP (plasmids list in *Table 2*). The allele expressing constitutively active Tor1 (TOR1-A1957V) was introduced in the *TOR1* genomic locus, replacing the endogenous *TOR1* copy. The strain used for the DNA circles propagation assay was described (*Shcheprova et al., 2008*), but we introduced in that background SPC42-CFP:kanMX4 and the Estradiol Binding Domain (EBD) fused to the GAL4 activator (EBD-GAL4:TRP1) (*Takahashi and Pryciak, 2008*). We backcrossed this strain five times into the S288C background. Medium was supplemented as indicated with Rapamycin at 200 ng/ml for 18–20 hr or with β-Estradiol at 200 ng/ml for 3 hr and glucose at 2%. All reagents were bought from Sigma-Aldrich (St. Louis, MO). The amino acid mixes, yeast nitrogen base and ammonium sulfate used to prepare Synthetic Dropout (SD) medium were purchased from FORMEDIUM (United Kingdom). The Agar was purchased from SERVA (Germany) and the yeast extract and the bactopeptone from BD (USA). Unless otherwise stated, all strains were grown on plates, at 30°C in YPD (Yeast extract, Peptone, Dextrose) medium. They were kept on the condition of interest (e.g. 30°C, 2% glucose) from the culture prepared the day before the experiment and then during the experiment itself.

### DNA circles propagation assay

Cells were grown overnight to low density on plates lacking uracil (-URA). The next morning cells still exponentially growing were streaked on YPD plates containing 1 μM β-Estradiol. β-Estradiol binds to the Estradiol Binding Domain (EBD) in EBD-Gal4, which consequently enters the nucleus and triggers the expression of a recombinase. This is responsible for recombination between the recombination sites (*Figure 1A*), thus excising the centromere from the DNA circle. After 4 hr in β-Estradiol,

**Table 1.** Yeast strains used in the study

| yYB number | Mating type | Genotype |
|---|---|---|
| 7828 | a | *nup49*::NUP49-GFP:HIS3; *his3Δ1 leu2Δ0 ura3Δ0 met15Δ0* |
| 4223 | a | *nup49*::NUP49-GFP:HIS3; *bud6*::natNT2; *his3Δ1 leu2Δ0 ura3Δ0 met15Δ0* |
| 10411 | a | *sec61*::SEC61-GFP:hpnNT1; *his3Δ0 leu2Δ0 met15Δ0 ura3Δ0* – clone 1 |
| 10412 | a | sec61::SEC61-GFP:hpnNT1; *his3Δ0 leu2Δ0 met15Δ0 ura3Δ0* – clone 2 |
| 10413 | a | sec61::SEC61-GFP:hpnNT1; *his3Δ0 leu2Δ0 met15Δ0 ura3Δ0* – clone 3 |
| 1879 | α | sec61::SEC61-GFP:TRP1; *shs1*::KAN; *his3Δ200 trp1Δ63 leu2Δ0 ura3-52 ade2-101 lys2-801* – clone1 |
| 1880 | a | sec61::SEC61-GFP:TRP1; *shs1*::KAN; *his3Δ200 trp1Δ63 leu2Δ0 ura3-52 ade2-101 lys2-801* – clone2 |
| 6770 | a | *nup49*::NUP49-GFP:HIS3; *tor1*::TOR1 A1975V: hpnNT1; *his3-Δ1 leu2Δ0 met15Δ0 ura3Δ0* |
| 4229 | a | *nup49*::NUP49-GFP:HIS3; *slt2*::kanMX4; *his3Δ1 leu2Δ0 ura3Δ0 met15Δ0* |
| 4231 | a | *nup49*::NUP49-GFP:HIS3; *slt2*::kanMX4; *his3Δ1 leu2Δ0 ura3Δ0 met15Δ0*<br>Plasmid: pYB 1273 (**Table 2**) carrying PKC1-R398P |
| 10622 | a | *nup49*::NUP49-GFP:HIS3; *bcy1*::hphNTI *his3Δ1 leu2Δ0 ura3Δ0 met15Δ0* – clone 1 |
| 10623 | a | *nup49*::NUP49-GFP:HIS3; *bcy1*::hphNTI *his3Δ1 leu2Δ0 ura3Δ0 met15Δ0* – clone 2 |
| 10624 | a | *nup49*::NUP49-GFP:HIS3; *bcy1*::hphNTI *his3Δ1 leu2Δ0 ura3Δ0 met15Δ0* – clone 3 |
| 4221 | a | *spc42*::SPC42-CFP:kanMX4; *trp1*::GAL4-EBD:TRP1; *leu2*::TETR-GFP:LEU2; *his3*::pGAL-REC:HIS3; *ura3-52 ade2-101 trp1-Δ63*<br>Plasmid: pPCM14 (224 tetO-REC-URA3-CEN-REC-LEU2) |
| 6099 | a | *spc42*::SPC42-CFP:kanMX4; *trp1*::GAL4-EBD:TRP1; *leu2*::TETR-GFP:LEU2; *his3*::pGAL-REC:HIS3; *ura3-52*<br>Plasmid: pPCM14 (224 tetO-REC-URA3-CEN-REC-LEU2) |
| 4222 | a | *spc42*::SPC42-CFP:kanMX4; *trp1*::GAL4-EBD:TRP1; *leu2*::TETR-GFP:LEU2; *his3*::pGAL-REC:HIS3; *bud6*::natNT2; *ura3-52 ade2-101 trp1-Δ63*<br>Plasmid: pPCM14 (224 tetO-REC-URA3-CEN-REC-LEU2) – clone 1 |
| 5547 | a | *spc42*::SPC42-CFP:kanMX4; *trp1*::GAL4-EBD:TRP1; *leu2*::TETR-GFP:LEU2; *his3*::pGAL-REC:HIS3; *bud6*::natNT2; *ura3-52 ade2-101 trp1-Δ63*<br>Plasmid: pPCM14 (224 tetO-REC-URA3-CEN-REC-LEU2) – clone 2 |
| 6521 | a | *spc42*::SPC42-CFP:kanMX4; *trp1*::GAL4-EBD:TRP1; *leu2*::TETR-GFP:LEU2; *his3*::pGAL-REC:HIS3; *bud6*::natNT2; *ura3-52*<br>Plasmid: pPCM14 (224 tetO-REC-URA3-CEN-REC-LEU2) – clone 1 |
| 6522 | a | *spc42*::SPC42-CFP:kanMX4; *trp1*::GAL4-EBD:TRP1; *leu2*::TETR-GFP:LEU2; *his3*::pGAL-REC:HIS3; *bud6*::natNT2; *ura3-52*<br>Plasmid: pPCM14 (224 tetO-REC-URA3-CEN-REC-LEU2) – clone 2 |
| 6523 | a | *spc42*::SPC42-CFP:kanMX4; *trp1*::GAL4-EBD:TRP1; *leu2*::TETR-GFP:LEU2; *his3*::pGAL-REC:HIS3; *bud6*::natNT2; *ura3-52*<br>Plasmid: pPCM14 (224 tetO-REC-URA3-CEN-REC-LEU2) – clone 3 |
| 7301 | α | *nsg1*::NSG1-GFP:HIS3; *spc42*::SPC42-CFP:kanMX4; *trp1*::GAL4-EBD:TRP1; *leu2*::TETR-GFP:LEU2; *his3*::pGAL-REC:HIS3; *ura3-52*<br>Plasmid: pPCM14 (224 tetO-REC-URA3-CEN-REC-LEU2) – clone 1 |
| 7302 | α | *nsg1*::NSG1-GFP:HIS3; *spc42*::SPC42-CFP:kanMX4; *trp1*::GAL4-EBD:TRP1; *leu2*::TETR-GFP:LEU2; *his3*::pGAL-REC:HIS3; *ura3-52*<br>Plasmid: pPCM14 (224 tetO-REC-URA3-CEN-REC-LEU2) – clone 2 |
| 5532 | a | *nsg1*::NSG1-GFP:HIS3; *his3Δ1 leu2Δ0 ura3Δ0 met15Δ0* |
| 6648 | a | *nup82*::NUP82-3sfGFP:kanMX4; $P_{URA3}$-TETR-mCherry:kanMX4; *spc42*::SPC42-yeGFP:hphNT1; *his3*::$P_{GAL}$-REC:HIS3; *trp1*::GAL4-EBD:TRP1; *ade2-101*;<br>Plasmid: pPCM14 (224 tetO-REC-URA3-CEN-REC-LEU2) – clone 1 |
| 6649 | a | *nup82*::NUP82-3sfGFP:kanMX4; $P_{URA3}$-TETR-mCherry:kanMX4; *spc42*::SPC42-yeGFP:hphNT1; *his3*::$P_{GAL}$-REC:HIS3; *trp1*::GAL4-EBD:TRP1; *ade2-101*;<br>Plasmid: pPCM14 (224 tetO-REC-URA3-CEN-REC-LEU2) – clone 2 |
| 6765 | a | *gcn5*::natNT2; *nup82*::NUP82-3sfGFP:kanMX4; $P_{URA3}$-TETR-mCherry:kanMX4; *spc42*::SPC42-yeGFP:hphNT1; *his3*::$P_{GAL}$-REC:HIS3; *trp1*::GAL4-EBD:TRP1; *ade2-101*;<br>Plasmid: pPCM14 (224 tetO-REC-URA3-CEN-REC-LEU2) – clone 1 |
| 6752 | a | *gcn5*::natNT2; *nup82*::NUP82-3sfGFP:kanMX4; $P_{URA3}$-TETR-mCherry:kanMX4; *spc42*::SPC42-yeGFP:hphNT1; *his3*::$P_{GAL}$-REC:HIS3; *trp1*::GAL4-EBD:TRP1; *ade2-101*;<br>Plasmid: pPCM14 (224 tetO-REC-URA3-CEN-REC-LEU2) – clone 2 |
| 3415 | a | *ade2*::hisG his3 leu2 lys2 ura3Δ0 trp1Δ63 hoΔ::$P_{SCW11}$-cre-EBD78-NatMX loxP-UBC9-loxP-LEU2 loxP-CDC20-Intron-loxP-HPHMX* |
| 3201 | a | *bud6*::kanMX4; *his3Δ1 leu2Δ0 met15Δ0 ura3Δ0* |
| 962 | a | *his3Δ1 leu2Δ0 met15Δ0 ura3Δ0* |
| 14902 | a | SPC42-CFP:kanMX4 *Gal4-EBD:TRP1 leu2*::LEU2 TetR-GFP *his*::HIS3 pGAL-Cre *ura3-52 trp1-Δ63 ADE2*::NatMX<br>Plasmid: pPCM14 (224 tetO-REC-URA3-CEN-REC-LEU2); pYB1273 |

DOI: https://doi.org/10.7554/eLife.28329.016

cells were suspended in medium lacking tryptophan (-TRP) and imaged with an Olympus BX50 microscope, equipped with a piezo motor, a monochromatic light source and a CCD camera (Andor885). The microscope was controlled with the TillVision software (Till Photonics/FEI Munich GmbH, Germany). Images were acquired with a 100x/1.4 NA oil immersion objective, $2 \times 2$ binning and 20 focal slices (0.3 μM spacing). Maximum intensity projection was used to localize the DNA circles and the SPBs. Cells in anaphase containing 1, 2 and 4 circles not co-localizing with the SPBs were analyzed for the presence of the circle in the mother and/or the bud. The DNA circle propagation frequency was calculated as:

$$1 - \left( \frac{n_1 \times p_1 + 2n_2 \times \sqrt{p_2} + 4n_4 \times \sqrt[4]{p_4}}{n_1 + 2n_2 + 4n_4} \right)$$

where $n1$, $n2$ and $n4$ are the number of cells respectively containing 1, 2 and 4 circles, and $p1$, $p2$ and $p3$ the percentage of cells retaining respectively 1, 2 or 4 circles in the mother. For the experiments in *Figures 2C* and *4E* and *Figure 2—figure supplement 1A* after 4 hr in β-Estradiol, cells were suspended in SD -TRP medium, immobilized on a cover slip with a SD -TRP pad (2% agar) and imaged using a Deltavision microscope (Applied Precision, Slovakia). The microscope was equipped with a CCD HQ$^2$ camera (Photometrics, Arizona), 250W Xenon lamps, Softworx software (Applied Precision, Slovakia) and a temperature chamber, set to the desired temperature. A 100x/ 1.40 NA U plan S Apochromat oil immersion objective (Olympus, Japan) was used. 30-min time lapse movies (1 frame per minute) with 20 stacks (0.25 μM spacing) were acquired. Maximum intensity projection was performed. The propagation flux of individual DNA circles through the bud neck in early and late anaphase was measured as a DNA circle passage frequency per minute, considering the moment when a circle is observed for the first time in the daughter compartment as the passage event.

## Anaphase duration and nuclear morphology analysis

For the data in *Figure 2* and *Figure 2—figure supplement 1*, cells were grown and samples prepared and imaged as explained in the previous paragraph. Early anaphase was defined as the time window starting with the entry of the nucleus into the bud and finishing with the formation of a dumbbell-shaped nucleus. Late anaphase corresponds to the further elongation of the nucleus, and particularly of the bridge between the two future daughter nuclei, and finishes with the resolution of that bridge at karyokinesis (*Figure 2—figure supplement 2A,B*). For the experiments shown in *Figure 2—figure supplement 2*, cells were grown overnight to low density on YPD plates. The next morning cells still exponentially growing, were suspended in -TRP medium and immobilized on a -TRP pad (2% agar). Cells were imaged with a Deltavision microscope as described in the previous paragraph. A $2 \times 2$ binning and an auxiliary 1.6x magnification were used. Time lapse movies of 25 s interval for 25 min and 7 stacks (0.3 μM spacing) were acquired. After 3D iterative deconvolution, neck width and M-D axis length were measured considering only cells where we could follow the entire nuclear division process. These measurements were performed in an equatorial focal section of each nucleus and plotted as mean distance over time.

**Table 2.** Plasmids used in the study

| pYB number | Backbone | Description |
| --- | --- | --- |
| 1309* | pRS315 | LEU2, CEN/ARS, HA3-TOR1-A1957V |
| 1273† | YCp50 | URA3, CEN, PKC1-R398P |
| 1274‡ | pRS316 | URA3, *Bck1-20* |
| 1316* | pRS315 | LEU2, CEN/ARS |

*originally described in *Reinke et al., (2006)*
†originally described in *Nonaka et al. (1995)*
‡originally described in *Helliwell et al. (1998)*
DOI: https://doi.org/10.7554/eLife.28329.017

## NPCs-DNA circles attachment assay

Cells were grown overnight as in 'DNA circles propagation assay'. The cells contained the model circle and expressed TetR fused to mCherry (TetR-mCherry), a Nup82 tagged with 3 copies of super folder GFP (Nup82-3x sfGFP) to label the NPCs and Spc42p with GFP (Spc42-GFP) to label the SPB. The cells were grown at either 30°C or 37°C. After 4 hr of β-Estradiol treatment, the cells were suspended in low fluorescent SD -TRP medium and immobilized on a cover slip with a SD -TRP pad (2% agar). For rapid imaging, a Nikon Eclipse T1 microscope was used, with acquisition times of 25 and 50 ms for GFP and mCherry respectively. The microscope was equipped with a LUDL BioPrecision2 stage with Piezo Focus, two 200 mW laser lines (DPSS 488 nm and DIode561 nm), a sCMOS camera (Orca Flash 4.0 V2) and a temperature chamber (Oko-lab), set to the desired temperature. A 100x/ 1.49 CFI Apochromat TIRF oil immersion objective was used. The microscope was controlled with the VisiVIEW software (Metamorph/Molecular Devices, California). For each channel (mCherry and GFP), 9 stacks (0.3 µM spacing) were acquired, with one bright field image in the center of the stack. Cells in anaphase containing one DNA circle dot distinct from the SPB were analyzed using the software Fiji (imagej.net/Fiji, *Schindelin et al., 2012*). 2 pixel wide GFP and mCherry fluorescence intensity profiles were measured for each cell along the nuclear rim (excluding the SPB). These measurements were performed in the focal plane for both Nup82-3x sfGFP and the DNA circle (tetR-mCherry). Cells where the DNA circle localized at, or close to the SPB were disregarded. All the single-cell Nup82-3x sfGFP traces were aligned relative to the corresponding brightest tetR-mCherry pixel (DNA circle) and averaged to obtain a mean profile, subsequently plotted for both channels. The mean Nup82-3x sfGFP intensity per position was only calculated for positions where the number of cells > 10. After background subtraction, the mean GFP intensity at the rim was set to 1. The GFP intensity at the tetR-mCherry peak was measured as fold induction compared to the normalized value of the rim.

## FLIP experiments

Cells were grown as explained in 'Anaphase duration and nuclear morphology analysis'. Time lapse movies of 3–5 s interval for 4–5 min were acquired. For the experiments shown in *Figures 4* and *5* and *Figure 5—figure supplements 1* and *2* (except *bcyΔ* mutant cells), cells were imaged with a confocal LSM 510 microscope, controlled by ZEN 2010 (Carl Zeiss Microimaging Inc, Germany). We used a Plan-Apochromat 63x/1.4 NA oil immersion objective and 3% of laser intensity with 25% laser output (Argon laser, 488 nm). Bleaching pulses were iterated (as indicated in the figures) for 80 times with 60% laser intensity. For *bcyΔ* mutant cells (*Figure 5* and *Figure 5—figure supplement 2*) all was as for *Figure 4* but the following: a confocal LSM 780 microscope (controlled by ZEN 2011, Carl Zeiss Microimaging Inc, Germany), 3.5% of laser intensity (laser output of 40%, Argon laser, 488 nm) and a multi-array 32PMT GaAsP detector were used. Bleaching pulses were iterated for 50 times with 100% laser intensity. For the experiments in *Figure 4—figure supplement 1*, everything was as for *bcyΔ* mutant cells but the following: 20% of laser intensity; bleaching pulses were iterated for 100 times with 100% laser intensity. For all FLIP experiments, bleaching pulses were iterated at every frame and quantification was performed using Fiji as follows: the total integrated fluorescence intensity was measured in the mother and bud compartments and in 3–5 neighboring control cells. After background subtraction, the fluorescence intensity of mother and bud compartments was normalized to the mean intensity of the control cells (to correct for fluorescent decay due to exposure) and set to 100%. The resulting single-cell fluorescence profiles were pooled to obtain a single profile. This was fit to a one phase decay function, using the software Prism 6 (GraphPad software, GraphPad Software, Inc., California). The initial Y0 value was constrained to 100%. The resulting best fit values for plateau and K and their relative errors were used to measure the Barrier Index (BI, see main text for BI definition). The standard error of the BIs was calculated by error propagation on the errors obtained from the fit. *bcy1Δ* mutant cells and a wild type strain were analyzed in parallel, using the LSM 780 microscope. This showed a significantly lower BI in *bcy1Δ* mutant cells. For comparison purposes, this BI was normalized relative to the wild type analyzed with the LSM 510 microscope. This allowed us to compare *bcy1Δ* with the experiments performed with the LSM 510 microscope and shown in *Figure 5*.

## Replicative lifespan analysis

Cells were streaked from −80°C and grown for 2 days on YPD plates. After 2 days they were streaked again on YPD plates and grown at either 30°C or 37°C overnight. The next morning they were streaked on fresh and pre-warmed YPD plates and grown for 2 hr at 30°C or 37°C. After 2 hr, virgin daughters were separated from mother-virgin daughter pairs and placed on defined spots on the plate, using a Zeiss Axioscope 40 microdissection microscope. The microscope was equipped with a 10X objective. Every 1.5 hr, the isolated daughters were visited and newly born daughters removed and counted. Dissection was performed for 10–14 hr per day and at room temperature for all experiments. Between dissection rounds, cells were kept in a wet box at either 30°C or 37°C. Overnight, the wet boxes was stored at +4°C, to slow down the cell cycle. Two independent experiments per condition with 20–40 virgin daughters per experiment were analyzed.

## Detection of DNA circles in aged cells by Southern blotting

A Southern blot was performed, as before (*Denoth-Lippuner et al., 2014*) but with DNA from young and aged cells both cultured at 30°C or 37°C. Young cells were harvested from an exponentially growing culture. Aged cells were purified according to the protocol of the mother enrichment program (*Lindstrom and Gottschling, 2009*), with some adaptations. Briefly, $5 \times 10^7$ cells were washed with PBS and labeled with Sulfo-NHS-LC-Biotin (Pierce/Thermo Fisher Scientific, Massachusetts) and recovered for 2 hr at 30°C or 37°C prior to the addition of β-Estradiol (1 μM final concentration). The cells were harvested after 26 hr of incubation at 30°C or 37°C. After a wash with PBS, batches of $2 \times 10^9$ cells were resuspended in 1 mL PBS supplemented with 50 μl streptavidin-coated magnetic beads (MicroMACS, Miltenyi Biotec, Germany), incubated for 30 min at 4°C and loaded onto LS MACS columns (Miltenyi Biotec) for purification. Cells were eluted using $1 \times$ PBS containing 2 mM EDTA and split into two fractions: (1) 10% of the cells were fixed with paraformaldehyde and the bud scars labeled with 5 μg/ml calcofluor white and visualized by microscopy. The fraction of aged cells in the population and the bud scar count per aged cells were quantified. (2) Cells were lysed and DNA was purified using standard methods. DNA content was quantified performing qPCR amplifying ACT1 from aged and young cells in triplicate. The fraction of DNA from aged cells per sample was equilibrated with DNA from young cells, to have in both samples same amount of DNA originating from aged cells. The amount of DNA loaded into the gel was equilibrated based on qPCR reads. A 0.6% agarose gel contained ethidium bromide and was run in TBE with ethidium bromide for 25 hr at 50 V at 4°C. The gel was blotted to a cationized nylon transfer membrane (Zeta-Probe GT, Bio-RAD, California) using standard protocols. Membranes were hybridized with a probe generated by conventional PCR, amplifying the rDNA locus of genomic DNA extracted from wild type yeast (*Lindstrom et al., 2011*) and 5' end-labeled with $^{32}$P. The blot was visualized using a Typhoon phosphoimager (GE healthcare, UK). The relative band intensity of the different ERC bands per sample were measured in Fiji (imagej.net/Fiji, *Schindelin et al., 2012*) and summed for plotting.

## Statistics

A two-tailed unpaired student's t-test was used to test for significance, for the nuclear morphology a two-way ANOVA followed by Tukey's multiple comparison test was used, for the replicative lifespan experiments a log-Rank (Mantel-Cox) test was used. For the anaphase durations, the non-parametric Mann-Whitney U test was used. Unless otherwise indicated, all the analyzed mutants/conditions were always compared to wild type cells (30°C, 2% glucose).

## Acknowledgements

We thank Fabrice Caudron, Juha Saarikangas, Ana-Maria Farcas, Marek Krzyzanowski, Xiuzhen Chen and Jette Lengefeld for critical reading of earlier versions of the manuscript, Anna Marzelliusardottir and Annina Denoth-Lippuner for technical support and former and present members of the lab for fruitful discussions over the years. We also thank the staff of the Institute of Biochemistry, the SCOPE-M staff for support in image acquisition and analysis and Michael Hall and Robbie Loewith for providing reagents.

## Additional information

### Funding

| Funder | Grant reference number | Author |
|---|---|---|
| Eidgenössische Technische Hochschule Zürich | | Yves Barral |
| European Research Council | BarrAge | Yves Barral |

The funders had no role in study design, data collection and interpretation, or the decision to submit the work for publication.

### Author contributions

Sandro Baldi, Alessio Bolognesi, Anne Cornelis Meinema, Data curation, Formal analysis, Investigation, Writing—original draft, Writing—review and editing; Yves Barral, Conceptualization, Formal analysis, Supervision, Funding acquisition, Project administration, Writing—review and editing

### Author ORCIDs

Alessio Bolognesi (iD) https://orcid.org/0000-0001-7268-4577
Anne Cornelis Meinema (iD) http://orcid.org/0000-0002-0002-3486
Yves Barral (iD) http://orcid.org/0000-0002-0989-3373

### Decision letter and Author response

Decision letter https://doi.org/10.7554/eLife.28329.020
Author response https://doi.org/10.7554/eLife.28329.021

## Additional files

### Supplementary files

• Transparent reporting form
DOI: https://doi.org/10.7554/eLife.28329.018

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
