## [Decision Letter]

Thank you for submitting your article "Heat stress promotes longevity through relaxing age confinement in the mother cell" for consideration by *eLife*. Your article has been reviewed by three peer reviewers, and the evaluation has been overseen by a Reviewing Editor and Naama Barkai as the Senior Editor. The reviewers have opted to remain anonymous.

The reviewers have discussed the reviews with one another and the Reviewing Editor has drafted this decision to help you prepare a revised submission.

Summary:

The authors report effects of heat stress, caloric restriction and the signaling pathways PKC, TOR & PKA on the retention of molecules in the mother nucleus. They suggests a model in which heat stress increases longevity as a result of DNA circle propagation to the bud and support this model by life-span experiments. All reviewers were enthusiastic about the study and support publication, provided that the authors address the following comments.

Essential revisions:

As you will see below, all three reviewers requested additional experimental support for the model (heat shock increases lifespan due to reduced retention of eccDNA). Alternatively, this conclusion should be toned down.

In addition, reference to previous studies should be made more complete.

Please respond to the other comments either by changes in the text, additional experiment if necessary and explanations to the reviewers.

Reviewer #1:

Baldi et al. present a solid set of experiments that demonstrate the effects of heat stress, caloric restriction and the signaling pathways PKC, TOR & PKA on the retention of molecules in the mother nucleus. They use DNA circles as a model system. Heat stress increases propagation of circles to daughters (and therefore, reduces retention in mothers), while caloric restriction reverses this effect. These nuclear retention measurements correlate well with longevity.

In general, the experiments support the conclusions. However, the last statement: correlation to longevity, is slightly weak. It would be strengthened by quantification of DNA circle daughter propagation in cells grown in 200 ng ml^-1^ rapamycin (we know that this condition increases lifespan – is propagation also increased?). How about analysis of PKC1-R398P in terms of DNA circles and lifespan at 2% and 0.1% glucose?

If any of this information is in the literature, the authors should state so.

If these experiments work out, I would suggest a different title: Heat stress promotes longevity in budding yeast by relaxing the confinement of age-promoting factors in the mother cell

It's helpful to identify the model organism – we don't know if these conclusions will extend to other organisms. This title is easier for me to understand.

Reviewer #2:

In the manuscript titled "Heat stress promotes longevity through relaxing age confinement in the mother cell", authors Barrel et al. reported heat stress results in weakened nuclear diffusion barriers, allowing increased distribution of extrachromosomal circular DNAs into the daughter cells. They found that this pathway is regulated by PKC via downstream Tor1 and Pkc1. Overall, the manuscript was well written and the experiments were thoughtfully designed and carefully executed. Most of the results are convincing and support their major conclusions. However, the manuscript feels incomplete without further addressing the following questions, especially considering the longevity focus of the manuscript title.

1) The authors showed that heat stress extended lifespan for mother cells by weakening the nuclear diffusion barrier. It is assumed that heat weakening the diffusion barrier will allow extrachromosomal rDNA circles (ERCs) to be segregated more into the daughter cells. This should be experimentally tested – whether heat stress results in less accumulation of ERCs in the mother cells.

2) Further, the propagation rate for model circular DNAs was 0.04 under normal conditions according Figure 1. This rate was increased to 0.17 under heat stress. Another way to read this data is that 96% of circles are retained in the mother and this rate decreases to 83% under heat stress. Is this change sufficient to significantly alter accumulated ERCs in old cells? Some mathematical estimates will help readers to understand and support the conclusion on longevity.

3) According to proposed model, like bud6 mutant, heat stress should also shorten lifespan for the daughter cells. This should also be experimentally tested and compared to bud6.

Reviewer #3:

The work by Yves Barral and coworkers represents a comprehensive analysis of the molecular and cell biological mechanisms for prolonged longevity in connection to heat stress in the budding yeast *Saccharomyces cerevisiae*. Barrel and coworkers describe how circular DNA evades retention in yeast mother cells by TOR and Pkc dependent pathways upon heat stress using advanced microscopic and molecular methods.

These observations are important for our understanding of the molecular mechanisms underlying aging. Though the conclusions are drawn for yeast, they may also apply for metazoans.

Substantive concerns

1) The experiment presented in Figure 8 suggests a model in which heat stress increases longevity as a result of DNA circle propagation to the bud. Two replicative lifespan experiments are shown in support. However, neither these nor other experiments in the manuscript show that reduced accumulation of circular DNA leads to increased replicative lifespan at 37°C. Hence the model is not supported. Either this model can be taken out of the manuscript or experiments must be made to support the model.

2) Introduction, paragraph five. To my knowledge, "This retention mechanism seems to be generic for DNA circles […] independent of their sequence" is not supported by the literature.

3) Subsection “DNA circles exchange between the mother and the daughter cells during early anaphase” final paragraph. The authors conclude "Thus the duration of anaphase cannot […] frequency with which circles propagate into daughter cells". I am a confused here because I interpret their experiments differently. A) Most circle transfer happen in early anaphase, B) early anaphase is shorter at 37°C and more circles propagate, C) early anaphase is longer under calorie restriction and less circles propagate. Hence there is correlation between anaphase and the frequency with which circles propagate, not the opposite, as stated – the manuscript should be clear on this point, since conclusions drawn from these experiments are repeated throughout the manuscript.

4) The Introduction contains a large section about aging factor in yeast that should be shortened into a few lines if at all mentioned (e.g. PMA1 retention and change in pH is not relevant). A review like this does not belong in a research paper, as it sidetracks the reader from the treated problem.

5) The result section describes a large number of control experiments. These control experiments are important for the main conclusion, but their description should be reduced substantially, put into supplementary results and/or some can be left out altogether. That way, important observations such as circle transfer being 6 times higher in early anaphase will also be more evident.

6) The discussion subsection “A “damage‐sharing” mechanism underlies the longevity effect of heat stress “– is well written though I have some concern about the argument "The idea that stress does not lead to […] one does not see really why cells should do better when they are stressed than when they are not" The argument that we humans cannot explain the connection between stress and lifetime extension does not exclude that this connection has evolved as result of selection.

7) Subsection “The diffusion barrier confines DNA circles to the mother cell during division in *S. cerevisiae*” paragraph two is not really a discussion. A summary, rather than a discussion, should be reduced.

---

## [Author Response]

Reviewer #1:Baldi et al. present a solid set of experiments that demonstrate the effects of heat stress, caloric restriction and the signaling pathways PKC, TOR & PKA on the retention of molecules in the mother nucleus. They use DNA circles as a model system. Heat stress increases propagation of circles to daughters (and therefore, reduces retention in mothers), while caloric restriction reverses this effect. These nuclear retention measurements correlate well with longevity.In general, the experiments support the conclusions. However, the last statement: correlation to longevity, is slightly weak. It would be strengthened by quantification of DNA circle daughter propagation in cells grown in 200 ng ml^-1^ rapamycin (we know that this condition increases lifespan – is propagation also increased?). How about analysis of PKC1-R398P in terms of DNA circles and lifespan at 2% and 0.1% glucose?

The analysis of the effect of the *PKC1-R398P* mutation on the retention of DNA circles has now been added to the manuscript. As is expected from our previous results the *PKC1-R398P* mutation does increase the propagation of the DNA circle to the daughter cells, in fitting with the fact that this mutation causes a decrease in barrier strength.

For what concerns the life-span of the *PKC1-R398P* mutant cells, we had already investigated this question. However, these cells are actually extremely sick, for probably a number of reasons that have nothing to do with the reduction of barrier strength. Accordingly, these cells are also very short-lived or easily killed by the dissection needle. Therefore, the experiment suggested by the reviewer cannot be carried out in a conclusive manner.

For what concerns rapamycin treatment, we assume that its effect in extending longevity must follow another path than barrier opening since this treatment increases the strength of the barrier rather than decreasing it. However, it is possible, if not even likely, that it acts by relaxing circles from retention in the mother cells through other mechanisms, such as possibly circle detachment from NPCs. Our data about DNA circle propagation (not shown) support the idea that rapamycin affects circle retention. However, since we have no data at this stage about how rapamycin affects circle retention and whether this is the mechanism by which it extends longevity, we have not added these data to this manuscript.

Altogether, these data indicate that indeed, calorie restriction and rapamycin treatment do not promote longevity through opening of the diffusion barrier, as our data already suggested. Furthermore, opening of the diffusion barrier, whether through growth at 37°C or in response to the *PKC1-R398P* mutation does indeed promote the propagation of the DNA circles to the daughter cells.

However, in order to strengthen our last statement, we have now added data showing that cells grown at 37°C accumulate 3.3 folds less DNA circles at the age of 16 than corresponding mother cells grown at 30°C.

If any of this information is in the literature, the authors should state so.

We went back through our manuscript and tried to make sure that we do not forget any relevant reference. However, in case we would have missed one or several papers that the reviewer has in mind, we would be very happy to go through them and cite them appropriately.

If these experiments work out, I would suggest a different title: Heat stress promotes longevity in budding yeast by relaxing the confinement of age-promoting factors in the mother cellIt's helpful to identify the model organism – we don't know if these conclusions will extend to other organisms. This title is easier for me to understand.

We thank the reviewer for this excellent suggestion.

Reviewer #2:[…] 1) The authors showed that heat stress extended lifespan for mother cells by weakening the nuclear diffusion barrier. It is assumed that heat weakening the diffusion barrier will allow extrachromosomal rDNA circles (ERCs) to be segregated more into the daughter cells. This should be experimentally tested – whether heat stress results in less accumulation of ERCs in the mother cells.

This is a very good point. To address it, we purified old mother cells from a population grown at 30°C and 37°C using the mother enrichment program. The ERC content was visualized and quantified using Southern-blot analysis. We observed a 3.3 fold lower ERC content in heat stressed aged cells, consistent with a heat-induced relaxation of the diffusion barrier strength.

2) Further, the propagation rate for model circular DNAs was 0.04 under normal conditions according Figure 1. This rate was increased to 0.17 under heat stress. Another way to read this data is that 96% of circles are retained in the mother and this rate decreases to 83% under heat stress. Is this change sufficient to significantly alter accumulated ERCs in old cells? Some mathematical estimates will help readers to understand and support the conclusion on longevity.

Mathematical studies have been carried out and are reported in at least two papers: Gillespie et al., 2004 and in Shcheprova et al., 2008 (Supplementary Figure 7). The mathematical modeling reported in the first study predicted that a retention probability above 0.99 is required to simulate experimentally obtained aging curves (Gillespie et al., 2004). The second study established that reducing retention even slightly has a big effect on longevity.

In the mathematical model, using a replication rate that is closer to the values that are estimated to be physiological and a retention rate of 83% (17% propagation), the DNA circles pass nearly as efficient as they replicate, resulting in marginal accumulation with age and decreased aging effects. These data are also reviewed in Denoth Lippuner et al., 2014. Moreover, we observed that heat stressed cells after 16 generations of aging showed a 3.3-fold reduction in ERC content with an 83% retention rate. Altogether, yes, a drop of 14% in retention is likely to have a strong effect on longevity.

We added these elements to the Discussion section in the manuscript.

3) According to proposed model, like bud6 mutant, heat stress should also shorten lifespan for the daughter cells. This should also be experimentally tested and compared to bud6.

Prompted by the reviewer’s comment, we started to do these experiments. We do see that heat stress shortens the life span of daughters. However, in these experiments we also see a defect at 30°C, such that the experiments are not conclusive. We have acquired evidence over the years that the rejuvenation of the daughters is very sensitive to experimental conditions (and the data in this paper certainly support this idea). For example, the duration of the shift to 4°C during the night and to which extend the plates dry during the experiment have strong impacts on rejuvenation of the daughters. We went out of time for these experiments and will not be able to satisfactorily address this point. However, we recognize that this is an important issue that needs to be addressed further in the future. We are currently investing increasing efforts in the development of microfluidics methods, and this will probably be the way to go in the future. Unfortunately, we are not yet able to catch chosen daughters in these experiments.

Reviewer #3:1) The experiment presented in Figure 8 suggests a model in which heat stress increases longevity as a result of DNA circle propagation to the bud. Two replicative lifespan experiments are shown in support. However, neither these nor other experiments in the manuscript show that reduced accumulation of circular DNA leads to increased replicative lifespan at 37°C. Hence the model is not supported. Either this model can be taken out of the manuscript or experiments must be made to support the model.

As explained above, we have now addressed this point by isolating old mother cells grown either at 30°C or at 37°C, using the mother enrichment program. We then quantified their ERC content using Southern-blot analysis. As explained above, these results indicate that cells grown at 37°C indeed accumulate less DNA circles as they age.

2) Introduction, paragraph five. To my knowledge, "This retention mechanism seems to be generic for DNA circles [...] independent of their sequence" is not supported by the literature.

The reviewer is actually correct: there are sequences that are able to relieve plasmids from efficient retention, such as the HMR and HML loci and other binding sites for transcription repressors. The data suggest that this acts by recruiting Sir2 and other histone deacetylases, possibly because they can counteract SAGA. However, this is out of the scope of this paper and we have therefore removed the erroneous statement without entering the topic any further.

3) Subsection “DNA circles exchange between the mother and the daughter cells during early anaphase” final paragraph. The authors conclude "Thus the duration of anaphase cannot […] frequency with which circles propagate into daughter cells". I am a confused here because I interpret their experiments differently. A) Most circle transfer happen in early anaphase, B) early anaphase is shorter at 37°C and more circles propagate, C) early anaphase is longer under calorie restriction and less circles propagate. Hence there is correlation between anaphase and the frequency with which circles propagate, not the opposite, as stated – the manuscript should be clear on this point, since conclusions drawn from these experiments are repeated throughout the manuscript.

This is indeed correct, we observe a correlation between the duration of anaphase and the retention of the circles in the mother cell. The shorter anaphase, the less the circles are retained. The point that we meant to make is that this correlation is opposite to the effects expected. A short anaphase should allow fewer plasmids to diffuse from the mother into the bud. We observe the opposite. Actually, we have observed in the past that cells with a diffusion barrier need longer to progress through anaphase. The data here suggest that the stronger the barrier, the longer the cells need. This would explain why they also pass fewer circles to the bud. Accordingly, barrier defects shorten the duration of anaphase. We believe that this reflects the constraints that the barrier imposes on the nuclear envelope during its elongation. Therefore, we believe that the correlation between anaphase duration and circle retention is indirect and reflects the effect of the diffusion barrier on both: impeding anaphase progression and preventing the passage of circles to the bud. We have corrected the text to eliminate our misleading statement. However, we did not feel that discussing the relationships between barrier strength and anaphase duration falls within the scope of this paper.

4) The Introduction contains a large section about aging factor in yeast that should be shortened into a few lines if at all mentioned (e.g. PMA1 retention and change in pH is not relevant). A review like this does not belong in a research paper, as it sidetracks the reader from the treated problem.

This is a fair point. We condensed the Introduction.

5) The result section describes a large number of control experiments. These control experiments are important for the main conclusion, but their description should be reduced substantially, put into supplementary results and/or some can be left out altogether. That way, important observations such as circle transfer being 6 times higher in early anaphase will also be more evident.

Thank you for this suggestion. The text has been adapted accordingly, and figures have been moved to supplement or merged with each other.

6) The discussion subsection “A “damage‐sharing” mechanism underlies the longevity effect of heat stress “is well written though I have some concern about the argument "The idea that stress does not lead to […] one does not see really why cells should do better when they are stressed than when they are not" The argument that we humans cannot explain the connection between stress and lifetime extension does not exclude that this connection has evolved as result of selection.

The reviewer is obviously correct, and we have softened our sentence accordingly. However, we would like to stress here that the intuition seems to not be too far-fetched since it is what we seem to observe: Stressed cells live longer not because they increase the fidelity with which they avoid or correct damage but essentially because they reduce the fidelity with which they confine it in the mother cell. The point that we want to introduce here is that this observation opens another question, which we think is even more important: Then, why the hell do cells confine these damages so efficiently to the mother cell if this has such adverse effects on their longevity? The idea that we are trying to introduce is that what the mothers retain might include beneficial elements. For example, DNA circles and protein aggregates could have adaptive roles. The barrier might serve the purpose of the mother in adapting to her environment. In case of stress, barrier opening could help share such beneficial elements between mother and bud. This discussion has been slightly expanded to bring the point more clearly across.

7) Subsection “The diffusion barrier confines DNA circles to the mother cell during division in S. cerevisiae” paragraph two is not really a discussion. A summary, rather than a discussion, should be reduced.

We have edited the text accordingly.